# TINYEYE: SHARPENING VISUAL REASONING OF TINY MODELS WITH OFFLINE POLICY OPTIMIZATION

## ABSTRACT

Multimodal reasoning with small vision–language models (VLMs) is increasingly important in real-world applications, yet their limited capacity makes optimization and alignment especially challenging. In this paper, we propose a holistic framework for offline policy optimization to sharpen the visual reasoning capabilities of small models. At its foundation is TinyEye-Data, a large-scale corpus of two million reasoning trajectories distilled from state-of-the-art VLMs across 68 verifiable tasks, which provides diverse and reliable binary supervision entirely in the offline setting. We instantiate the framework through a four-stage pipeline: (1) native-resolution warm-up for robust vision–language alignment, (2) instruction tuning on TinyEye-Data to establish a broad reasoning foundation, (3) annealed rejection sampling to mine hard cases and refine supervision, and (4) Discriminative Direct Preference Optimization (DDPO), a new margin-based objective that formulates policy learning as reward classification and resolves the likelihood displacement issues of DPO. Stages (3) and (4) together form the core of verifiable offline reinforcement learning, where rejection sampling refines signals and DDPO optimizes the policy against them. The resulting model, TinyEye-2B, achieves state-of-the-art results across diverse reasoning benchmarks, reaching 50.3% on MMMU, 55.2% on MathVerse, and 63.9% on HallBench, outperforming other models of comparable scale by significant margins.

## 1 INTRODUCTION

Post-training has become a cornerstone in the development of visual–language models (VLMs), enabling them to follow instructions and perform long chain-of-thought reasoning. Supervised fine-tuning (SFT) with curated instruction following datasets is among the earliest and most widely adopted approaches (Liu et al., 2023; Li et al., 2024a; Chen et al., 2023; Bai et al., 2023; 2025; Li et al., 2025b). More recently, the community has witnessed a paradigm shift beyond SFT toward methods that directly incorporate reward or preference signals. Direct Preference Optimization (Rafailov et al., 2023; Zhang et al., 2024b) (DPO) aligns models by contrasting preferred and non-preferred responses, while reinforcement learning (RL) methods such as PPO (Schulman et al., 2017), GRPO (Shao et al., 2024), and GSPO (Zheng et al., 2025) explicitly optimize model policies against reward feedback obtained from real-time sampled rollouts. Together, these three families of techniques define the core toolkit for post-training VLMs.

Despite these successes, such approaches also face inherent limitations, especially under constraints on model parameter size. Supervised fine-tuning (SFT) relies solely on positive examples and ignores negative evidence, which can lead to overfitting and a collapse in output diversity (Li et al., 2024b), particularly when the model's capacity is limited (Kaplan et al., 2020; Hoffmann et al., 2022). Direct Preference Optimization (DPO) avoids explicit reward modeling but suffers from likelihood displacement (Cho et al., 2025; Yang et al., 2025b): the additive margin in its loss may suppress the probability of negative responses but fails to sufficiently reinforce positive ones (Cho et al., 2025), sometimes even penalizing desirable behaviors (Pal et al., 2024a). Reinforcement learning (RL) methods, in contrast, provide stronger optimization signals but come with high computational costs, poor sample efficiency due to limited rollout reuse, and training instability that makes them difficult to tune. When reinforcement signals are high-variance and unstable, capacity-limited LLMs are especially prone to entropy collapse (Cui et al., 2025; Le et al., 2025), often resulting in low-diversity, repetitive outputs or even endless tandem loops during on-policy training. Recent

studies (Shumailov et al., 2024; Wenger, 2024) also indicate that if synthetic or self-generated data are reused recursively without careful filtering, models tend to first forget rare, long-tail knowledge and eventually distort even common pretrained knowledge—a process known as model collapse, which degrades both the realism and diversity of the model's capabilities.

Multimodal reasoning with small models has many appealing real world applications requiring edge deployment. However, the above drawbacks are particularly pronounced in such settings, where naive reinforcement learning often exploits superficial shortcuts (Sarch et al., 2025; Xia et al., 2025) rather than learning grounded reasoning, leading to poor generalization. DPO is also highly sensitive to data quality (Xu et al., 2024): low-margin or noisy preference pairs can weaken parameter updates, and additive margin inside its loss objective may further suppress the probability of positive responses (Pal et al., 2024a), resulting in degraded and unstable alignment. For this reason, filtering such pairs is strongly advised (Morimura et al., 2024). Altogether, these observations underscore the urgent need for an alignment method that is discriminative, stable under limited capacity, and efficient without relying on costly on-policy rollouts.

We propose a unified offline policy optimization framework that elicits strong multimodal reasoning capabilities in compact visual–language models. The framework contains three integral components: large-scale offline distillation from diverse reasoning trajectories, offline rejection sampling to mine and refine supervision signals, and a final stage of offline policy optimization through Discriminative Direct Preference Optimization (DDPO). Central to this design is TinyEye-Data, a verifiable dataset that enables broad coverage for distillation and provides the foundation for learning entirely in the offline setting. Together, these components form a coherent pipeline: distillation supplies rich and diverse reasoning signals, rejection sampling sharpens them into high-quality positive–negative supervision, and DDPO optimizes over the curated pairs with a margin-based discriminative objective. This coupling ensures that each stage reinforces the next, yielding stable and scalable alignment without relying on unstable on-policy rollouts.

We instantiate the framework through a four-stage pipeline as shown in Figure 1. In Stage-1, a native-resolution warm-up adapts a SigLIP2 (Tschannen et al., 2025) backbone into a ViT variant capable of flexible variable-resolution processing and robust vision–language alignment. In Stage-2, we conduct large-scale instruction tuning over TinyEye-Data, incorporating both curated multimodal corpora and trajectories distilled from strong teacher models such as MiMo-VL (Yue et al., 2025) and Keye-VL (Team et al., 2025c), thereby establishing a broad reasoning foundation in an offline manner. In Stage-3, we apply annealed rejection sampling to mine hard cases, filter degenerate outputs, and amplify supervision density through an additional round of tuning. Finally, in Stage-4, we introduce DDPO, a discriminative objective that enforces a margin-based update rule. In the binary reward setting common to multimodal reasoning, DDPO guarantees that gradient descent always increases the probability of preferred responses while suppressing negatives, thus resolving the likelihood displacement issue of DPO (Razin et al., 2025) and providing stable optimization.

In conclusion, our work makes three main contributions:

- We propose a holistic framework for **offline policy optimization** that integrates large-scale distillation, rejection sampling, and discriminative policy optimization, enabling stable reasoning alignment for small multimodal models.

- We construct **TinyEye-Data**, a large-scale verifiable corpus of two million reasoning trajectories distilled from diverse teacher models across 68 tasks.

- We introduce **DDPO**, a discriminative policy optimization objective that provably guarantees monotonic reinforcement of positives while suppressing negatives, directly resolving the likelihood displacement (Cho et al., 2025; Yang et al., 2025b) issue of DPO.

- We demonstrate strong empirical performance: our model, **TinyEye-2B**, achieves state-of-the-art results on multimodal and math reasoning benchmarks, surpassing strong open-weight baselines such as InternVL3-2B (Zhu et al., 2025) and Ovis2-2B (Lu et al., 2024).

## 2 METHOD

In this section, we introduce the proposed framework. We first introduce TinyEye-Data, a curated dataset that emphasizes verifiable supervision and multi-teacher distillation to provide both relia-

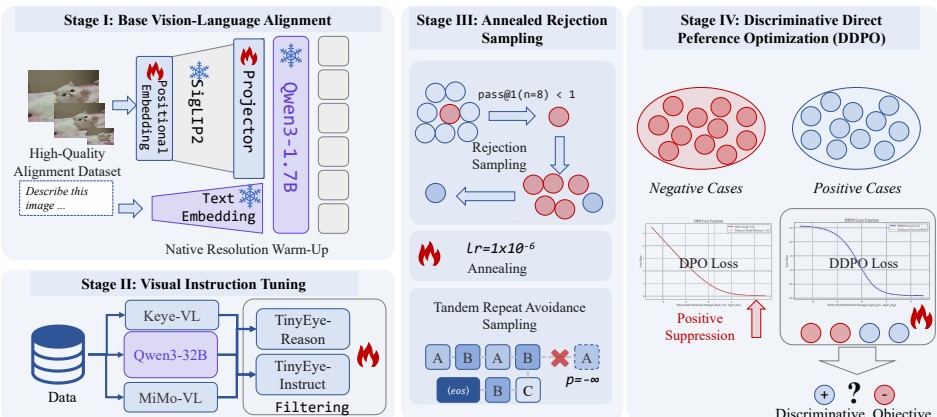

Figure 1: An overview of the proposed offline policy optimization pipeline.

bility and broad coverage across multimodal reasoning tasks. Building on this dataset, TinyEye is trained through the instantiated four-stage pipeline as illustrated in Figure 1.

## 2.1 TINYEYE-DATA

Effective offline policy optimization requires diverse and verifiable reasoning data: diversity ensures that compact models encounter a broad spectrum of tasks, while verifiability guarantees stable signals for subsequent rejection sampling and discriminative optimization. To this end, we construct TinyEye-Data, a heterogeneous pool spanning text, vision, mathematics, STEM, puzzles, and documents, designed to expose small models to both core skills and long-tail reasoning challenges.

We aim to enable robust visual reasoning under limited model capacities. Our core strategy is to construct compact and verifiable supervision through a base–tail paradigm: a solid base of essential skills complemented by a highly diverse tail of long-tail tasks. This design follows Li et al. (2025b) on the principle of diversity first, then quality, where we deliberately maximize coverage of reasoning styles and domains before applying strict distillation and cleansing. By combining multi-teacher offline distillation with curated verification signals, TinyEye-Data equips small models with reliable supervision while mitigating noise and collapse risks at the source.

**Data construction.** Instead of indiscriminately aggregating large-scale instructions, we curate a dataset pool called TinyEye-Data comprising 179 datasets across 14 categories. Collection proceeds along two complementary lines: TinyEye-Instruct and TinyEye-Reason, both following the base–tail philosophy.

**TinyEye-Instruct** provides the foundation for multimodal instruction tuning. Its base consolidates key skills through math and geometry (17 datasets), chart and table reasoning (11), and scientific diagrams (11). The tail then pushes diversity by incorporating long-tail tasks: 27 perception VQA datasets, 11 OCR and document-understanding datasets, and 4 captioning corpora, along with 5 text-only QA datasets to stabilize instruction following. In total, TinyEye-Instruct integrates 86 datasets spanning mathematics, structured data, diagrams, perception, documents, captioning, and text. Quality is safeguarded with filtering strategies such as majority-vote distillation on AOPS-Instruct (Mahdavi et al., 2025). A detailed breakdown is shown in Table 6.

**TinyEye-Reason** extends the same diversity-first design with explicit emphasis on reasoning trajectories. Its base consists of canonical math corpora (AOPS-Instruct, MATH, AIME, etc.), while the tail greatly broadens coverage: 10 game and puzzle datasets, 11 STEM domains (chemistry, medicine, finance, philosophy, etc.), and 38 general VQA datasets grounding reasoning in visual understanding. Multimodal reasoning is further strengthened by 17 math-related datasets, 7 puzzle-style datasets, and 5 STEM-oriented corpora requiring joint symbolic–visual reasoning. Some datasets by themselves contain highly diverse sub-domains. For instance, StackExchange-longtail (Lambert et al., 2023) contains 2 million longtail examples covering 171 topics. In total, TinyEye-Reason covers 93 datasets across seven categories, ranging from Olympiad-level proofs to pediatric medicine and counterfactual history. The composition is summarized in Table 7.

Some datasets naturally serve both instruction tuning and reasoning distillation. Across the pool, supervision signals fall into two types: verifiable targets (equations, multiple-choice) strictly validated via exact matching, and natural-language targets judged by VLM-as-a-Judge to retain expressive flexibility. This layered design ensures both verifiability and diversity, enabling compact models to internalize stable reasoning patterns without sacrificing breadth of coverage.

**Multi-teacher offline distillation.** Building on this supervised dataset, we further distill high-quality reasoning trajectories. Relying on a single teacher model could often produce noisy trajectories that "accidentally" or "luckily retry" lead to the correct answer. Accordingly, we assign teachers by task type: MiMo-VL-7B-RL for math-related visual questions, Keye-VL-8B-Preview for other VQA tasks, and Qwen3-32B for text-only prompts. For each question, we generate 8 independent trajectories with temperature sampling—the 8 trajectories allow us to compute pass@8 as a statistical measure of dataset difficulty, which can further guide the subsequent training stages.

**Reward Labeling.** To guarantee the quality of distilled trajectories, we adopt task-specific reward surrogates. For equations and multiple-choice problems, we apply `math-verify` Kydlíček to perform symbolic and numerical equivalence checks, providing a near noise-free supervision signal. For natural-language responses, we use a lightweight VLM-as-Judge based on Qwen2.5-VL-3B-Instruct (Wang et al., 2024b; Team, 2025), which can jointly interpret image and text contexts to assess semantic consistency. This prevents cases where an answer is factually correct but phrased inconsistently, and it reduces the risk of introducing image–text misalignment hallucinations during subsequent training. In this way, we enforce strict correctness on verifiable tasks while ensuring semantic robustness on open-ended ones, thereby improving the reliability and controllability of the distilled corpus.

**Cleaning and aggregation.** To block degenerate supervision signal, we apply two key post-distillation steps. First, we remove long, low-variation tandem repeats using a linear-time detector (Algorithm 2), preventing the model from imitating collapse-like patterns present in teacher outputs. Second, among the remaining candidates we perform shortest-chain aggregation, keeping for each prompt only the shortest valid chain of thought. This strategy instills an explicit preference for efficient reasoning without sacrificing correctness, and attenuates exhaustive or enumerative traces.

After the above construction and filtering, the resulting corpus is overall more compact, contains substantially fewer repetitive patterns, and preserves higher sample-level diversity, making it better aligned with the limited learning capacity of small models.

## 2.2 VISION-LANGUAGE ALIGNMENT AND INSTRUCTION TUNING

**Stage-1: Vision-Language Alignment.** To build a base model for our task, we consider a native resolution VLM where we adopt *Native Resolution Warm-Up* to adapt the vision encoder to varied input sizes. The proposed process performs vision-language alignment with a frozen language backbone. In particular, we use Qwen3-1.7B (Yang et al., 2025a) as the language backbone and the SigLIP2-So400M (Tschannen et al., 2025) vision encoder, which has a modified $14 \times 14$ patch size (originally $16 \times 16$) and a fixed input resolution of $448 \times 448$. To enable the native resolution capabilities, we represent each input image as a patch sequence $V \in \mathbb{R}^{N \times d}$. The absolute positional embeddings $P_{\text{abs}}$ are interpolated to the target grid size $(H, W)$, yielding $\tilde{P} \in \mathbb{R}^{N \times d}$. Following Keye-VL (Team et al., 2025c), we apply a learnable two-dimensional rotary positional embedding (RoPE) layer with parameters $\phi$:

$$V' = \text{RoPE}_\phi(V + \tilde{P}). \tag{1}$$

Unlike previous approaches that continue pretraining the vision tower independently with large-scale contrastive learning, we warm up the ViT to handle native-resolution inputs while simultaneously performing vision–language alignment. During this stage, both the vision encoder and the LLM remain frozen, while only the projector module and the learnable positional encoding layers are trained. We use ALLaVA (Chen et al., 2024a) and LLaVA-ReCap-CC3M (Li et al., 2024a) datasets, and optimize with a standard cross-entropy loss for captioning and question answering.

**Stage-2: Instruction Tuning.** Building upon the aligned base model, we perform large-scale multimodal instruction tuning to equip the model with robust task-following and reasoning capabilities. Training is conducted on the curated TinyEye-Instruct and TinyEye-Reason datasets, which combine broad task coverage with compact and verifiable supervision across image captioning, visual question answering, mathematical reasoning, and general knowledge tasks.

We optimize the model with the standard autoregressive cross-entropy loss over the instruction–response pairs. Given a sequence of target tokens $y = \{y_1, \ldots, y_T\}$ conditioned on multimodal input image tokens $I$ and text tokens $x$, the objective is:

$$\mathcal{L}_{\text{CE}} = -\sum_{t=1}^{T} \log p_\theta(y_t \mid y_{<t}, I, x), \tag{2}$$

where $p_\theta$ denotes the model's conditional probability distribution. This stage transitions the model from merely aligned representations to effective instruction following, thereby establishing a stable foundation for subsequent hard-case mining and policy optimization.

## 2.3 VERIFIABLE OFFLINE REINFORCEMENT LEARNING

**Stage-3: Annealing Rejection Sampling Fine-Tuning.** In Stage 3, we shift the training focus from saturated prompts to more informative borderline or hard cases. Concretely, we estimate the teacher's success rate for each prompt as $\text{pass@1}_{(n=8)}(x)$. Prompts with $\text{pass@1}_{(n=8)}(x) = 1$ are considered saturated and removed, while the remaining subset $\mathcal{H} = \{x : \text{pass@1}_{(n=8)}(x) < 1\}$ defines the hard set for annealed resampling. On $\mathcal{H}$, we sample $K = 4$ candidates per prompt, thereby concentrating supervision on outputs that lie closer to the error boundary and thus carry richer alignment signal.

When generating such candidates, small-capacity models are prone to collapse into low-entropy tandem repetitions (e.g., endlessly concatenating short fragments). To mitigate this, we introduce Tandem Repeat Avoidance Sampling (TRAS). During decoding, given the partial sequence $y_{1:t}$, a linear-time detector $D(\cdot)$ identifies newly formed concatenated repeats. Once detected, the offending tokens are added to the banned set $B_{t+1}$, which is passed to a `LogitsProcessor` interface of Transformers library to mask degenerate continuations. To maintain efficiency, we store recent repeat patterns in an LRU buffer and only perform prefix matching against this buffer, dynamically refreshing recency. In this way, TRAS explicitly disrupts nascent repetition loops while preserving normal decoding. The pseudo-code of TRAS can be found in Algorithm 3.

After candidate generation, we score outputs using the same reward annotation method as in Stage 2 and apply the principle of Rejection Sampling Fine-Tuning (RFT). Specifically, only high-quality "positive" samples, as judged by the reward annotation method, are retained for supervised fine-tuning, while low-scoring "negative" samples are discarded. The underlying rationale is that by repeatedly generate-filter-finetune, the policy distribution of model is gradually shifted from the policy of base model toward a reward-aligned distribution. Compared with on-policy reinforcement learning methods that penalize negative samples, RFT better accommodates the fragility of small models, offering greater stability, simpler optimization, and reduced risks from directly using noisy negative feedback.

During fine-tuning, we further anneal the learning rate and train solely on positives from $\mathcal{H}$, ensuring stable convergence on genuinely challenging cases. By discarding saturated prompts and curating preference pairs nearer to the decision boundary, this stage delivers a tempered supervision signal: it suppresses degenerate repetitions, promotes exploration without collapse, and steers the student model more reliably toward alignment objectives. The retained positive–negative pairs also serve as valuable inputs for the discriminative optimization in Stage 4.

**Stage-4: Discriminative Direct Preference Optimization.** Since the reward signal in our task is inherently binary, which means each response to a given prompt is labeled either correct or incorrect, we aim to tailor the contrastive learning objective of DPO such that optimization directly focuses on the binary classification of reward labels. To this end, we reformulate the original DPO's preference modeling problem into an explicit binary discrimination task: the model must reliably classify responses into correct or incorrect categories, making the objective more consistent with the supervision structure of binary rewards in visual reasoning tasks. Motivated by this, we propose *Discriminative Direct Preference Optimization (DDPO)*, which reinterprets the relative likelihood differences in DPO as binary classification probabilities, and optimizes an interval-based objective designed for this two-class reward setting.

We first define the binary classification probability for a single response as

$$r(x, y) = \sigma\big(\Delta(x, y)\big), \qquad \Delta(x, y) = \ell_\theta(y|x) - \ell_{\text{ref}}(y|x), \tag{3}$$

where $\ell_\theta(y|x)$ is the length-normalized log-likelihood under the current policy and $\ell_{\text{ref}}(y|x)$ is the log-likelihood under a reference model. Through the sigmoid mapping, $\Delta(x, y)$ is interpreted as the probability that the response is correct.

For a preference pair $(y^+, y^-)$, the margin signal is given by the probability difference

$$d = \sigma\big(\Delta(x, y^+)\big) - \sigma\big(\Delta(x, y^-)\big). \tag{4}$$

A natural way to optimize this margin is with a hinge loss

$$L_{\text{hinge}}(\theta) = \max(0, 1 - d). \tag{5}$$

To enable smooth gradient-based optimization, we adopt the log-sigmoid as a differentiable surrogate, yielding the Discriminative Direct Preference Optimization (DDPO) objective

$$L_{\text{DDPO}}(\theta) = -\log \sigma\big(\alpha d\big), \tag{6}$$

where $\alpha > 0$ controls the sharpness of the margin. This formulation establishes a binary discriminative training objective, the policy is thus trained to decide, for each prompt, which response should be assigned the positive reward label and which the negative one. By introducing the sigmoid smoothing, DDPO avoids the likelihood displacement present in standard DPO, while providing sign-preserving gradient signals for both positives and negatives. From a perspective of sequence-level policy optimization, DDPO objective can view as a offline surrogate of GSPO method which concentrate on optimizing margin of advantages[1]. In practice, this stage consolidates the supervision: the model not only imitates high-quality responses but also learns to reliably separate them from poor ones, achieving more robust alignment under binary reward settings.

## 3 EXPERIMENTS

### 3.1 EXPERIMENT SETTINGS

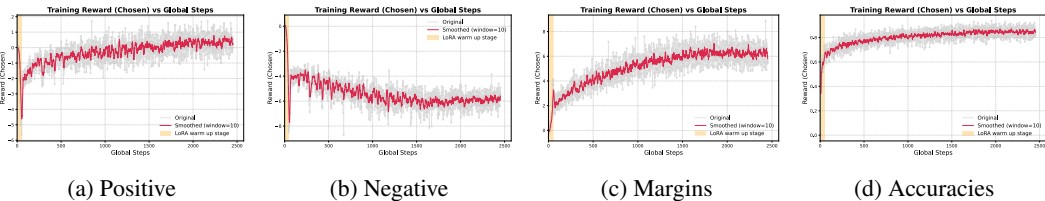

|     (a) Positive     |     (b) Negative     |     (c) Margins     |     (d) Accuracies     |

Figure 2: Train rewards v.s. global steps. A 10-step moving average is used to visualize. Yellow areas are warm-up stages of random initialized parameters of LoRA layers.

**Training Details.** For hyperparameters of training, please refer to Appendix C. Particularly, as illustrated in Figure 2, the reward dynamics during LoRA-based DDPO training exhibit a clear separation between positive and negative samples. After the warm-up of random initialized parameters of LoRA layers, the reward for positive examples (Figure 2a) steadily increases and stabilizes over time, indicating that the policy consistently strengthens its preference for high-quality responses. In contrast, the reward for negative examples (Figure 2b) progressively decreases, reflecting the model's ability to suppress low-quality responses. Consequently, the reward margin between positives and negatives (Figure 2c) grows throughout training, demonstrating that DDPO not only imitates preferred behaviors but also enlarges their separation from undesirable ones. This trend is further confirmed by the accuracy curve (Figure 2d), which rises toward saturation, showing that the policy becomes increasingly reliable in distinguishing good from poor responses.

**Benchmarks.** We evaluate our 2B model across three main tasks to show its multiple capability: multimodal reasoning, textual reasoning and general multimodal question answering. For multimodal reasoning, we choose MMMU (val) (Yue et al., 2023), MathVision (Wang et al., 2024a),

---

[1]For proof, please refer to Appendix B.

| Models | MMMU(Val) | MathVision | MathVista | MathVerse | LogicVista | WeMath |
|---|---|---|---|---|---|---|
| **TinyEye-2B-Stage4 (Ours)** | 50.3 | **32.6** | 67.9 | **55.2** | **41.7** | **52.4** |
| Ovis2-2B (Lu et al., 2024) | 45.6 | 17.7 | 64.1 | 29.4 | 34.7 | 9.9 |
| InternVL3-2B (Zhu et al., 2025) | 48.6 | 21.7 | 57.0 | 25.3 | 36.9 | 22.4 |
| Qwen2.5-VL-3B (Bai et al., 2025) | **51.2** | 21.9 | 61.2 | 31.2 | 40.3 | 22.9 |
| Ovis2-4B (Lu et al., 2024) | 49.0 | 21.5 | **69.6** | 38.5 | 35.3 | 16.9 |
| GPT-5 (OpenAI, 2025) | 81.8 | 72.0 | 81.9 | 81.2 | 70.0 | 71.1 |
| Gemini-2.5-Pro (DeepMind, 2025) | 74.7 | 69.1 | 80.9 | 76.9 | 73.8 | 78.0 |
| GLM-4.5V-108B (Team et al., 2025a) | 75.4 | 65.6 | 84.6 | 72.1 | 62.4 | 68.8 |
| Qwen2.5-VL-72B (Bai et al., 2025) | 68.2 | 39.3 | 74.2 | 47.3 | 55.7 | 49.1 |
| InternVL3-78B (Zhu et al., 2025) | 72.2 | 43.1 | 79.0 | 51.0 | 55.9 | 46.1 |

Table 1: Main results on multimodal reasoning datasets. We choose the vision-only part of Math-Verse. Some results are from InternVL-series (Zhu et al., 2025) technical report and OpenCompass (Contributors, 2023) platform. We **highlight** the best performance and underline the second best of smaller VLMs in this table.

| Models | AIME24 | AIME25 | MATH500 | GPQA-D | MMLU-Pro |
|---|---|---|---|---|---|
| **TinyEye-2B-Stage4 (Ours)** | 33.3 | 36.7 | 88.0 | 43.9 | 58.5 |
| GLM-4.1V-9B (Team et al., 2025a) | 36.2 | 32.0 | 81.8 | 50.3 | 57.1 |
| Kimi-VL-2506-16B (Team et al., 2025d) | 54.0 | 39.1 | 91.8 | 42.3 | 68.5 |
| GLM-4.5V-106B (Team et al., 2025a) | 72.8 | 80.1 | 94.2 | 56.6 | 69.7 |
| Qwen2.5-VL-72B (Bai et al., 2025) | 15.0 | 13.3 | 82.8 | 49.0 | 71.2 |
| GPT-5 (OpenAI, 2025) | 90.0 | 94.6 | 97.8 | 85.7 | 85.6 |
| Gemini-2.5-Pro (DeepMind, 2025) | 92.0 | 88.0 | 95.2 | 86.4 | 84.1 |
| Grok-4 (xAI, 2025) | - | 90.0 | 96.2 | 88.1 | 85.3 |
| Claude-Sonnet-4 (Anthropic, 2025) | - | 76.3 | 93.8 | 74.5 | 83.8 |

Table 2: Main results on textual math reasoning tasks. We apply some VLMs evaluated on these benchmarks, and use the score from their technical report.

MathVista (Lu et al., 2023), MathVerse (Zhang et al., 2024a), LogicVista (Xiao et al., 2024) and WeMath (Qiao et al., 2024). For textual reasoning, we select AIME24&AIME25 (math-ai, 2024; OpenCompass, 2025), MATH500 (Hendrycks et al., 2021), GPQA-Diamond (Rein et al., 2023) and MMLU-pro (Wang et al., 2024c). For general multimodal question answering tasks, we select MMBench (MMB) (Liu et al., 2024b), MMStar (Chen et al., 2024b), MMVet (Yu et al., 2023), AI2D (Kembhavi et al., 2016), HallBench (Guan et al., 2024) and BLINK (Fu et al., 2024).

**Baselines.** Several VLMs of the similar scales (2B-4B) are selected to compare with our model. Ovis2-2B, 4B (Lu et al., 2024), InternVL3-2B (Zhu et al., 2025) and Qwen2.5-VL-3B (Bai et al., 2025) are chosen. We also report the scores of some advanced VLMs like GPT-series (OpenAI, 2025; Hurst et al., 2024) and Gemini-series (DeepMind, 2025). Details are in experiment tables.

## 3.2 EVALUATION RESULTS

**Results on multimodal reasoning benchmarks.** Despite having only 2B parameters, our model, TinyEye-2B-Stage4 achieves strong performance across a wide range of multimodal reasoning benchmarks. It nearly outperforms all existing models of similar or smaller scale and even rivaling much larger ones. As shown in Table 1, TinyEye-2B-Stage4 achieves 55.2 on MathVerse, significantly surpassing other similar-scale models. It is also the only lightweight model to exceed 30 on MathVision (32.6) and sets a new state-of-the-art (SOTA) among small-scale VLMs on WeMath with a score of 52.4, outperforming models with up to 40× more parameters, such as InternVL3-78B and GLM-4.5V-108B. These results demonstrate that TinyEye-2B-Stage4 not only closes the performance gap with large-scale VLMs but also offers a highly efficient and scalable solution for multimodal reasoning tasks under resource constraints.

**Results on textual reasoning benchmarks.** Evaluation results are displayed in Table 2. Here we report VLMs with 9B parameters or more with official results. Our model achieves strong performance across several textual reasoning benchmarks. It reaches 88.0 on MATH500, outperforming larger VLMs like GLM-4.1V-9B (81.8) and Qwen2.5-VL-72B (82.8). On AIME25, it surpasses GLM-4.1V-9B by +4.7 points. It also performs competitively on GPQA-D (43.9) and MMLU-Pro

| Models | MMBench | MMStar | MMVet | AI2D | HallBench | BLINK |
|---|---|---|---|---|---|---|
| **TinyEye-2B-Stage4 (Ours)** | 81.5 | 61.3 | **73.0** | 79.7 | **63.9** | 44.3 |
| Ovis2-2B Lu et al. (2024) | 78.6 | 56.7 | 58.3 | 82.7 | 50.2 | 47.9 |
| InternVL3-2B Zhu et al. (2025) | 80.8 | 60.7 | 62.2 | 78.7 | 42.5 | 50.3 |
| Qwen2.5-VL-3B Bai et al. (2025) | 79.1 | 55.9 | 63.2 | 81.6 | 46.3 | 47.6 |
| Ovis2-4B Lu et al. (2024) | **83.6** | **61.9** | 65.5 | **85.7** | 53.8 | **53.0** |
| GPT4o Hurst et al. (2024) | 82.1 | 64.7 | 69.1 | 84.6 | 55.0 | 68.0 |
| Claude3.5 Sonnet Anthropic (2024) | 83.4 | 65.1 | 70.1 | 81.2 | 55.2 | - |
| Gemini-2-flash DeepMind (2024) | 83.0 | 69.4 | - | - | - | - |
| GLM-4.5V-108B Team et al. (2025a) | 88.2 | 75.3 | 75.2 | 93.7 | 64.5 | 65.3 |
| Qwen2.5-VL-72B Bai et al. (2025) | 88.3 | 70.8 | 76.2 | 88.4 | 55.2 | 64.4 |
| InternVL3-78B Zhu et al. (2025) | 88.8 | 72.5 | 81.3 | 96.0 | 59.1 | 66.3 |

Table 3: Main results on multimodal general datasets. Some results are from InternVL-series (Zhu et al., 2025) technical report and OpenCompass (Contributors, 2023) platform. We **highlight** the best performance and underline the second best of smaller VLMs in this table.

| Method | MathVista | MathVision | MATH500 | Hallbench |
|---|---|---|---|---|
| TinyEye-2B-Stage2 | 63.8 | 28.7 | 86.7 | 35.9 |
| TinyEye-2B-Stage3 | 66.0 | 30.4 | 87.2 | 48.7 |
| **TinyEye-2B-Stage4 (DDPO)** | 67.9 | **32.6** | **88.0** | **63.9** |
| TinyEye-2B-Stage3 + DPO | **68.0** | 31.0 | 87.4 | 55.8 |

Table 4: Performance of TinyEye-2B at different training stages and with policy optimization methods (DPO/DDPO) on four benchmarks.

(58.5), outperforming several larger models. These results depict that our lightweight model can match or even exceed the performance of much larger models on textual reasoning tasks.

**Results on general multimodal benchmarks.** On general multimodal benchmarks, TinyEye-2B-Stage4 achieves performance close to or better than models twice its size. For example, it outperforms Ovis2-2B and InternVL3-2B across almost all benchmarks. It also outperforms Ovis2-4B on HallBench, and illustrates close results on MMBench and MMStar. These results reveal that our 2B model can deliver competitive general understanding and reasoning capabilities, comparable to 4B-level VLMs.

**Case Study.** For examples of case study, please refer to Appendix F.

## 3.3 ABLATION STUDY

**Ablation on training stages.** We conduct an ablation study across different training stages. As shown in Table 4, model performance improves progressively at each stage. For example, on Hall-Bench, accuracy rises from 35.9% after instruction fine-tuning (Stage 2) to 48.7% with hard-case rejection sampling (Stage 3). Similar improvements are observed on other benchmarks, such as MathVista (63.8% → 66.0%) and MathVision (28.7% → 30.4%). These results indicate that carefully designed training stages can steadily enhance the model's reasoning ability across diverse mathematical benchmarks.

**Ablation on policy optimization.** We further analyze the effect of preference-based policy optimization. As shown in Table 4, Incorporating DPO in default setting of `trl` library after Stage 3 improves MathVista from 66.0% to 68.0% and HallBench from 48.7% to 55.8%. More notably, applying DDPO at Stage 4 leads to consistent and significant improvements, e.g., HallBench accuracy surges to 63.9% (a 28-point gain over Stage 2), and the model achieves the best performance on MathVision (32.6%) and MATH500 (88.0%). These results highlight that DDPO is particularly effective for binary classification tasks like HallBench, while also contributing to robust gains across a variety of mathematical datasets.

**Comparison on thinking and non-thinking modes** To assess the impact of enabling thinking mode, we perform an ablation study by comparing model performance with and without it. Specifically, we append "no think" to the prompt to disable this mode. As shown in Table 5, the absence of thinking leads to clear degradation on all benchmarks: MathVista drops by 12.2 points, MathVi-

| Method | MathVista | MathVision | MATH500 |
|---|---|---|---|
| **TinyEye-2B-Stage4** (thinking) | 67.9 | 32.6 | 88.0 |
| **TinyEye-2B-Stage4** (no thinking) | 55.7 | 19.7 | 77.4 |

Table 5: Think v.s. no think mode.

sion by 12.9, and MATH500 by 10.6. These consistent declines indicate that structured reasoning plays a crucial role in guiding VLMs toward correct solutions, and that thinking mode significantly enhances the model's ability to tackle reasoning-intensive tasks.

## 4 RELATED WORKS

**Variants of DPO.** DPO aligns models by maximizing the log-likelihood margin between preferred and non-preferred samples, but raises issues of reference dependence, likelihood displacement, margin design, gradient stability, and noise robustness. Variants address these in different ways: SimPO (Meng et al., 2024) weakens reliance on reference models with a simplified objective and fixed margin; Alpha-DPO (Wu et al., 2025) introduces instance-adaptive margins to balance alignment and diversity; DPO-Positive (Pal et al., 2024b) enforces "only increase" updates for preferred samples; Refined-DPO (Gallego, 2024) improve noise tolerance via importance weighting; ORPO (Hong et al., 2024) uses win–loss ratios for single-stage alignment.

**Offline policy optimizations in VLMs Training.** In vision and multimodal settings, V-DPO (Xie et al., 2024) curbs hallucinations through image–text consistency, while MIA-DPO (Liu et al., 2024c) leverages multi-image inputs to mitigate data scarcity. Methods such as VideoDPO (Liu et al., 2024a), VistaDPO (Huang et al., 2025), and TEMPO (Li et al., 2025a) incorporate temporal and semantic constraints for video generation and understanding. VLM applications include LLaVA-Reasoner-DPO (Zhang et al., 2024c) for reasoning, CLIP-DPO (Ouali et al., 2024) for hallucination reduction. At scale, InternVL (Chen et al., 2024c) and Keye-VL (Team et al., 2025e) integrate mixed preference optimization with large multimodal preference datasets, while MiniCPM-V (Yao et al., 2024) applies DPO to learn from AI feedback and improve factual consistency in visual understanding.

**Thinking VLMs.** With the development of VLMs, prior works develop thinking and reasoning abilities of them for solving more complex tasks. Recent advances in multimodal reasoning have shifted from static instruction-tuning toward reasoning-centric training and adaptive inference. MiMo-VL introduces a four-stage pretraining pipeline with mixed on-policy reinforcement learning to enhance chain-of-thought (CoT) capabilities (Yue et al., 2025). For video reasoning, Keye-VL employs a five-mode cold-start mixture to regulate when and how models should think, while Keye-VL 1.5 integrates Slow-Fast encoding and long-context tuning for temporal understanding (Team et al., 2025c;b). GLM-4.1V-Thinking applies reinforcement learning with curriculum sampling (RLCS) to a 9B VLM (Team et al., 2025a). Meanwhile, GPT-5 adopts a unified routed system, where a lightweight model defers to a deeper "thinking" pathway when complexity demands it (OpenAI, 2025).

## 5 CONCLUSION

In this work, we propose a unified offline policy optimization framework that equips compact visual–language models with strong multimodal reasoning capabilities. At its core is TinyEye-Data, a large-scale verifiable corpus distilled from diverse teacher models, which supports a four-stage training pipeline combining native-resolution warm-up, large-scale distillation, annealed rejection sampling, and Discriminative Direct Preference Optimization (DDPO). DDPO resolves the likelihood displacement issue of DPO by provably reinforcing positives while suppressing negatives, ensuring stable offline optimization. Our 2B-parameter model, TinyEye-2B, achieves state-of-the-art performance on multimodal and math reasoning benchmarks, surpassing strong open-weight baselines such as InternVL3-2B and Ovis2-2B.

## 6 ETHICS STATEMENT

This work does not involve any human subjects or sensitive personal data. The usage of all datasets strictly complies with their respective licenses.

Our methods are intended solely for academic and scientific purposes. We do not foresee direct harmful applications, but acknowledge that misuse could occur if applied without proper safeguards. We encourage responsible use of the research outcomes, with attention to fairness, transparency, and legal compliance.

## 7 REPRODUCIBILITY STATEMENT

We have taken several measures to ensure the reproducibility of our work. All details of the proposed model, preprocessing steps of datasets and algorithms with full hyperparameter settings and training procedures provided are described in the main text. Proofs of theoretical results are given in Appendix B.

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

## A  THE USE OF LARGE LANGUAGE MODELS (LLMS)

In this work, LLMs were employed in four main ways: (i) to aid and polish the writing for clarity and style; (ii) to assist in retrieval and discovery of related work; (iii) to support research ideation by suggesting possible directions and problem formulations; and (iv) to provide coding assistance, including code generation, debugging, and optimization suggestions.

All outputs were critically reviewed, verified, and refined by the authors. The core scientific ideas, methodology, and contributions remain entirely the authors' own.

## B  PROOF OF DISCRIMINATIVE DIRECT PREFERENCE OPTIMIZATION

Given a LLM policy $\pi_\theta$ to be optimized, there exists prompt $x$ and a generated candidate sequence $y$, we denote the length-normalized log-probability as

$$\ell_\theta(y|x) = \frac{1}{|y|} \sum_{t=1}^{|y|} \log \pi_\theta(y_t \mid x, y_{<t}),$$

for a sampled group $\{y_i\}_{i=1}^G$, each sequence receives a reward $r_i = r(x, y_i)$, which is then normalized within the group as

$$\hat{A}_i = \frac{r_i - \bar{r}}{s}, \qquad \bar{r} = \frac{1}{G} \sum_{j=1}^G r_j, \quad s = \sqrt{\frac{1}{G} \sum_{j=1}^G (r_j - \bar{r})^2}.$$

The GSPO objective is given by

$$J_{\text{GSPO}}(\theta) = \mathbb{E}_{x, \{y_i\} \sim \pi_{\theta_{\text{old}}}} \left[ \frac{1}{G} \sum_{i=1}^G \min \left( s_i(\theta)\hat{A}_i, \; \text{clip}(s_i(\theta), 1 - \varepsilon, 1 + \varepsilon)\hat{A}_i \right) \right],$$

where $s_i(\theta) = \exp(\ell_\theta(y_i|x) - \ell_{\theta_{\text{old}}}(y_i|x))$ denotes the sequence-level importance ratio.

Specifically, we consider a case of group size is restricted to two and reward values are binary, i.e., $G = 2$, and $r_i \in \{0, 1\}$, containing a positive instance $y^+$ and a negative instance $y^-$ with corresponding rewards $r^+$ and $r^-$, the normalized advantage difference simplifies to

$$\hat{A}^+ - \hat{A}^- = \frac{r^+ - r^-}{s}, \qquad s = \frac{|r^+ - r^-|}{2},$$

which further reduces to

$$\hat{A}^+ - \hat{A}^- = 2 \frac{r^+ - r^-}{|r^+ - r^-|} = 2 \, \text{sign}(r^+ - r^-).$$

Thus, in our binary case, the advantage difference degenerates to the **sign of the reward difference**.

Since the sign function is non-differentiable, we introduce a smooth approximation:

$$\text{sign}(z) \approx 2\sigma(\alpha z) - 1, \qquad \sigma(z) = \frac{1}{1 + e^{-z}}, \; \alpha > 0.$$

Consequently,

$$\hat{A}^+ - \hat{A}^- \approx 4\sigma(\alpha(r^+ - r^-)) - 2.$$

Up to scaling constants, which do not affect the optimum in expectation, we obtain the core approximation:

$$\hat{A}^+ - \hat{A}^- \propto \sigma(\alpha(r^+ - r^-)).$$

To connect rewards with the policy, we adopt the concept of *language models are reward models* from DPO algorithm. To reflect the binary nature of the reward, we use a sigmoid function to map the log-ratio between the policy and the reference model into the binary classification domain:

$$r(x, y) = \sigma(\Delta(x, y)), \qquad \Delta(x, y) = \ell_\theta(y|x) - \ell_{\text{ref}}(y|x).$$

Here, the reward is interpreted as the probability of correctness, modeled as a sigmoid transformation of the log-ratio between the policy and a reference model. The margin between positive and negative sequences is then

$$d = r^+ - r^- = \sigma(\Delta^+) - \sigma(\Delta^-).$$

Then, we can reformulated the GSPO objective as

$$J_{\text{GSPO}}(\theta) = \mathbb{E}\left[(\hat{A}^+ - \hat{A}^-) \cdot f(\theta)\right].$$

In our reformulation, the GSPO objective naturally connects to the margin-based classification paradigm underlying support vector machines (SVMs). Specifically, the difference term $\hat{A}^+ - \hat{A}^-$ functions as a margin signal that determines the correct ordering between $y^+$ and $y^-$. In the classical SVM framework, such a margin is enforced through the hinge loss,

$$\ell_{\text{hinge}}(y^+, y^-) = \max\big(0, 1 - (\hat{A}^+ - \hat{A}^-)\big),$$

which penalizes violations of the desired ranking $y^+ \succ y^-$. However, the hinge loss is non-smooth and thus less amenable to gradient-based optimization in large-scale neural models. To address this, we adopt a smooth surrogate analogous to replacing the hinge loss with a softplus/logistic formulation, yielding

$$L_{\text{DDPO}}(\theta) = -\log \sigma(\alpha(\sigma(\Delta^+) - \sigma(\Delta^-))).$$

This objective defines **Discriminative Direct Preference Optimization** (DDPO).

Let's see the case of Direct Preference Optimization (DPO), the objective is defined as

$$\mathcal{L}_{\text{DPO}}(\pi_\theta; \pi_{\text{ref}}) = -\mathbb{E}_{(x,y_w,y_l)\sim\mathcal{D}}\left[\log \sigma\big(\beta(\Delta^+ - \Delta^-)\big)\right],$$

where $\Delta^+ = \log \frac{\pi_\theta(y_w|x)}{\pi_{\text{ref}}(y_w|x)}$ and $\Delta^- = \log \frac{\pi_\theta(y_l|x)}{\pi_{\text{ref}}(y_l|x)}$. The gradients with respect to the transformed logits are

$$\frac{\partial L}{\partial \Delta^+} = -\beta(1 - \sigma(\beta(\Delta^+ - \Delta^-))), \quad \frac{\partial L}{\partial \Delta^-} = \beta(1 - \sigma(\beta(\Delta^+ - \Delta^-))).$$

While the expressions exhibit a superficially consistent sign pattern—negative for the preferred response and positive for the dispreferred response—the directional signal is only preserved relative to the difference $(\Delta^+ - \Delta^-)$. Specifically, the objective is invariant under additive shifts, i.e., for any constant $c$, replacing $\Delta^+ \mapsto \Delta^+ + c$ and $\Delta^- \mapsto \Delta^- + c$ leaves both the loss and its gradients unchanged. This additive shift invariance implies that the optimization dynamics depend solely on relative differences and are insensitive to the absolute scale of the logits.

Such invariance introduces non-identifiability in the learned implicit reward: the model cannot distinguish between reward functions that differ by a constant offset. In practice, this can lead to gradient degeneracy, particularly in binary reward settings. For example, simultaneously increasing or decreasing both $\Delta^+$ and $\Delta^-$ has no effect on the loss, producing vanishing or ambiguous gradient signals and potentially stalling optimization. Thus, although the partial derivatives of DPO appear to have fixed signs, they do not exhibit true sign-preserving behavior in an absolute sense, but only in a relative one.

By contrast, Discriminative Direct Preference Optimization (DDPO) introduces a nonlinear transformation prior to differencing, yielding the objective

$$L_{\text{DDPO}}(\theta) = -\log \sigma\big(\alpha(\sigma(\Delta^+) - \sigma(\Delta^-))\big),$$

with gradients

$$\frac{\partial L}{\partial \Delta^+} = -\alpha\, \sigma'(\Delta^+)\, \sigma(-\alpha d), \quad \frac{\partial L}{\partial \Delta^-} = \alpha\, \sigma'(\Delta^-)\, \sigma(-\alpha d),$$

where $d = \sigma(\Delta^+) - \sigma(\Delta^-)$. Since $\sigma'(\cdot) > 0$ and $\sigma(-\alpha d) > 0$, the gradient signs are strictly preserved—$\Delta^+$ is always pushed upward, and $\Delta^-$ always downward—independent of additive shifts. This design eliminates the degeneracy observed in DPO and guarantees stable, non-degenerate updates even under binary reward signals.

## C  TRAINING DETAILS

All experiments were conducted on 4 machines, each equipped with 8 A100-80GB GPUs, for a total of 32 GPUs. The training framework leveraged **Accelerate** with **DeepSpeed ZeRO-2**, employing **bf16 mixed precision**, **FlashAttention-2**, and distributed data parallelism. Unless otherwise specified, the maximum input length was set to 8K tokens, and **neat packing** was enabled to improve batch efficiency. The optimizer used was **AdamW**, with a **cosine decay** learning rate schedule, a warmup ratio of 0.1, and gradient clipping set to 1.0. The per-GPU batch size was 1, with gradient accumulation over 4 steps, resulting in an effective global batch size of 128. The specific configurations for each training stage are as follows:

- **Stage 1 (Vision–Language Alignment):** The language model and vision encoder were frozen, while only the projection layer and positional encodings were trained for 1 epoch. The maximum sequence length was 8K, and the learning rate was set to 5e-5.

- **Stage 2 (Instruction Fine-tuning):** All parameters were unfrozen and trained for 1 epoch with a learning rate of 5e-5. Gradient checkpointing was enabled.

- **Stage 3 (Annealed Rejection Sampling Fine-tuning):** Training was conducted on the hard-case subset $\mathcal{H}$, with the sequence length extended to 16K. The learning rate was reduced to 5e-6 for annealing, while other configurations remained consistent with the previous stages.

- **Stage 4 (DDPO):** LoRA-based parameter-efficient fine-tuning was applied to the projection layer and feed-forward layers. $\alpha$ of DDPO set to 1.0. The frozen base model was used as the reference policy. Training lasted 1 epoch with a learning rate of 5e-5, a maximum sequence length of 8K, and the use of memory-optimized gradient checkpointing from Unsloth.

| Category | Datasets |
|---|---|
| Math & Geometry QA | CLEVR-Math, GEOS, GeoQA, Geometry3K, IconQA, MapQA, Super-CLEVR, TabMWP, UniGeo, Geo170k, geo3k, GeomVerse, MM-MathInstruct, MMathCoT, multimath-300k, geoqa-plus, gqa |
| Chart & Table QA | ChartQA, Chart2Text, DVQA, HiTab, LRV-Chart, LRV-Normal-filtered, TabMWP, ROBUT-WTQ, ROBUT-WikiSQL, ROBUT-SQA, FigureQA |
| Diagram & Science QA | AI2D, Diagram-Image-to-Text, ScienceQA, Infographic-VQA, Infographic, InterGPS, Raven, MMSci, Gaokao-Datasets, K12-VQA, MM-Chem-Exam |
| General & Perception VQA | A-OKVQA, Visual7W, VizWiz, ST-VQA, TallyQA, VSR, Screen2Words, VisText, VisualMRC, VQA-RAD, SROIE, PMC-VQA, Infographic-VQA-LLaVA, HateMemes, WebSight, GUI-Grounding, ShareGPT4o, ShareGPT4V-COCO, ShareGPT4V-Knowledge, ShareGPT4V-LLaVA, ShareGPT4V-SAM, LLaVA-Wild-12K, LLaVA-Wild-39K, LLaVAR-GPT4-20K, VisionFlan-filtered, Mulberry, VisualWebInstruct |
| Document & OCR | Chrome-Writting, ORAND-Car-A, UReader-CAP, UReader-IE, Docmatix-Multi-20k-en, TextOCR, TextCaps, IAM, IIIT5K, Rendered-Text, HME100K |
| Image Caption for Pretrain | Allava-Instruct-VFlan, Allava-Instruct-Laion, Allava-Instruct-Chinese, LLaVA-ReCap-CC3M |
| Pure-text QA | WildChat, MegaScience, Hermes3, ToolAce, AOPS-Instruct-filtered |

Table 6: Composition of the **TinyEye-Instruct** dataset pool. We organize datasets into categories including math & geometry QA, chart & table QA, diagram & science QA, General & Perception VQA , Document & OCR, Image Caption for Pretrain, and Pure-text QA.

| Category | Datasets |
|---|---|
| Pure-text Math | AOPS-Instruct-filtered, OpenThoughts, Calculus, MATH, AIME1983-2023 |
| Pure-text Game & Puzzle | 24Points, Chinese-chess, Chinese-go, Japanese-shogi, Chinese-riddle, English-riddle, corporate-governance-reasoning, RelatLogic-Reasoning, Reasoning-gym, Stocks |
| Pure-text STEM | Chemistry-molecule-MCQ, Chemistry-reaction-MCQ, StackExchange-longtail, MegaScience, WildSci, Philosophy-Ethics-Morality, quantitative-finance-reasoning, Pediatric-medical-reasoning, counterfactual-history-reasoning, academic-chain, OlympiadExtended |
| General VQA | chartqa, CLEVR, coco, docvqa, ocr-vqa, pisc, sam, share-textvqa, sqa, textvqa, vg, web-celebrity, web-landmark, wikiart, A-OKVQA, multi-hiertt, DVQA, infovqa, IRV-chart, plotQA, PMC-VQA, TQA, VizWiz, VQA-AS, VQA-RAD, VQA2.0, chart2Text, CharQA, RoBUT-SQA, sharegpt4o, sharegpt4v, visual7w, ArxivQA, EXAMS-V, GVLQA, OpenSpaces, Spacellava, VisualWebInstruct, FigureQA |
| Multi-modal Math | CLEVR-Math, GEOS, GeoQA, Geometry3K, IconQA, MapQA, Super-CLEVR, TabMWP, UniGeo, Geo170k, geo3k, GeomVerse, MM-MathInstruct, MMathCoT, multimath-300k, geoqa-plus, gqa |
| Multi-modal Game & Puzzle | VisualPuzzles, seekworld, PuzzleVQA, AlgoPuzzleVQA, GameQA, VisualSphinx, Zebra-CoT |
| Multi-modal STEM | scienceqa, ai2d, MMK12, MM- Chem-Exam, K12-VQA |

Table 7: Composition of the **TinyEye-Reason** dataset pool. We organize datasets into categories including Pure-text Math, Pure-text STEM, Pure-text STEM, General VQA, Multi-modal Math, Multi-modal Game & Puzzle, and Multi-modal STEM.

## D    DATASET DETAILS

Here we list the details of datasets we use in Table 6 and Table 7.

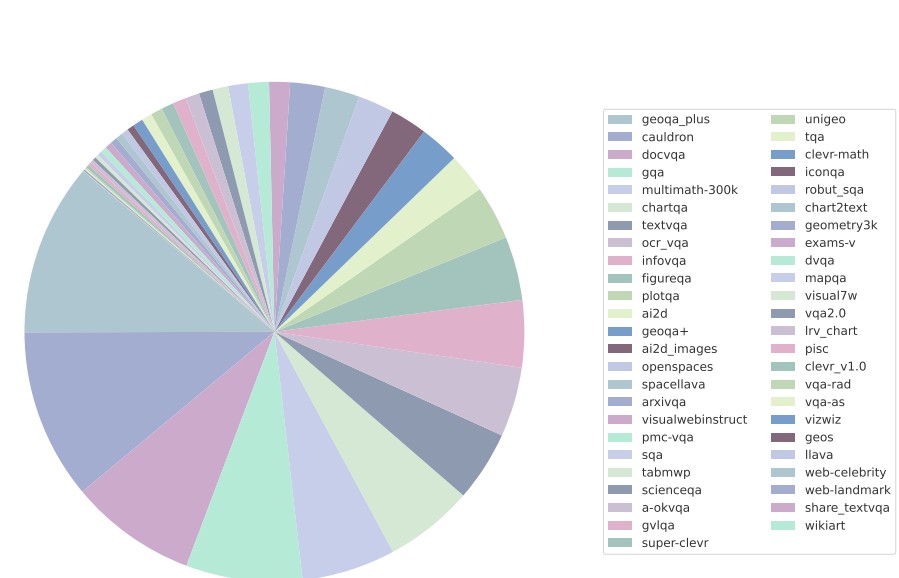

Figure 3: The distribution of the category of General VQA datasets inside **TinyEye-Reason**

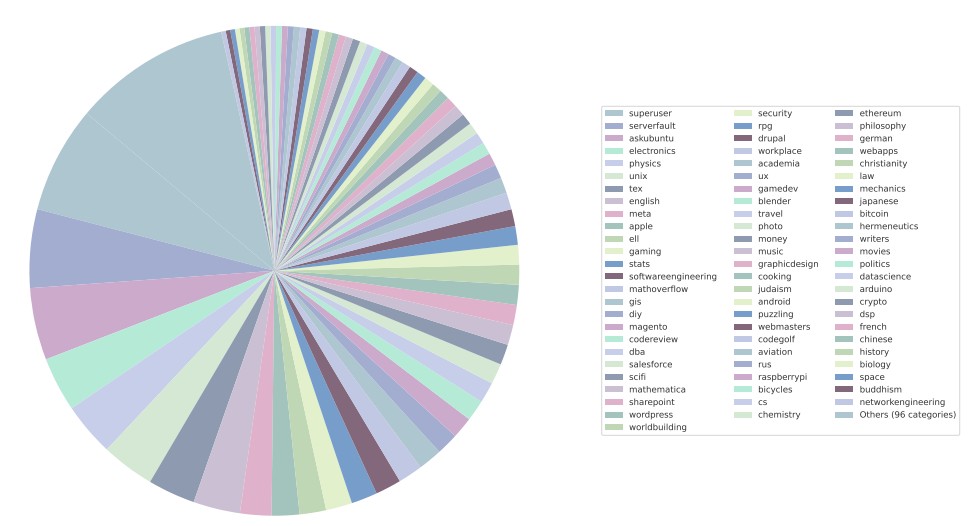

Figure 4: The distribution of the topics of longtail domain QA from StackExchange-longtail dataset inside **TinyEye-Reason**

# E    PSEUDO-CODES

---

**Algorithm 1:** ISTANDEMREPEATATK$(S, k, n, m)$

---

**Input:** string $S$ of length $L$, step $k \in \mathbb{N}$, repeats $n \geq 2$, probes $m$
**Output: True/False**

**if** $k < 1$ **or** $n < 2$ **or** $m < 1$ **or** $m > k - 1$ **or** $k > \lfloor L/n \rfloor$ **then return** *False*;
$J \leftarrow$ SAMPLEWITHOUTREPLACEMENT$\big(\{1, 2, \ldots, k-1\}, m\big)$;
**foreach** $j \in J$ **do**
    $v \leftarrow S[L - j]$ **for** $i \leftarrow 1$ **to** $n - 1$ **do**
        **if** $S[L - j - k \cdot i] \neq v$ **then**
            $\llcorner$ **return** *False*

**return** *True*

---

**Algorithm 2:** ISTANDEMREPEAT$(S, k_{\min}, n, m)$

---

**Input:** string $S$ of length $L$, min step $k_{\min}$, repeats $n \geq 2$, probes $m$
**Output: True/False**

$K_{\max} \leftarrow \left\lfloor \dfrac{L}{n} \right\rfloor$;
**for** $k \leftarrow k_{\min}$ **to** $K_{\max} - 1$ **do**
    **if** ISTANDEMREPEATATK$(S, k, n, m)$ **then**
        $\llcorner$ **return** *True*
**return** *False*

---

---

**Algorithm 3:** Tandem Repeat Avoidance Sampling with LRU Mechanism

---

**Input:** Current token sequence $seq$, logits $scores$
**Output:** Updated logits $scores$
$p \leftarrow$ DETECTTANDEMREPEATTAIL$(seq, k_{\min}, k_{\max}, n)$;
**if** $p \neq \emptyset$ **then**
    **if** $p \notin$ *LRU buffer* **then**
        insert $p$ into LRU buffer (as most recent);
        **if** *size(LRU buffer) > max capacity* **then**
            evict least-recently-used pattern from LRU buffer;
    **else**
        move $p$ to most recent position in LRU buffer;

$B \leftarrow \emptyset$ ;                               `// tokens to ban`
**foreach** *pattern p in LRU buffer* **do**
    $m \leftarrow$ LONGESTPREFIXMATCHLEN$(p, seq)$;
    **if** $m > 0$ **then**
        move $p$ to most recent position in LRU buffer ;        `// refresh recency`
    **if** $0 < m < |p|$ **then**
        $B \leftarrow B \cup \{p[m]\}$ ;           `// ban next token of prefix`
    **if** $m = |p|$ **then**
        $B \leftarrow B \cup \{p[0]\}$ ;               `// ban restart head`

**if** $B \neq \emptyset$ **then**
    **if** *top-k constraint enabled* **then**
        $B \leftarrow B \cap$ TOPK$(scores)$
    **if** $B \neq \emptyset$ **then**
        **if** *hard ban* **then**
            $scores[B] \leftarrow -\infty$
        **else**
            $scores[B] \leftarrow scores[B] - \lambda$

**return** $scores$;

---

# F   CASE STUDY

9. 下列四个图形中，线段 *BE* 是 △*ABC* 的高的是（  ）

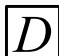

*Slove this problem.*

<think>
The current task for the user is to determine which figure shows that line segment BE is the height of triangle ABC. First, recall the definition of a height in a triangle: the height is the perpendicular segment drawn from a vertex to the opposite side (or its extension), with the foot of the perpendicular being on the opposite side. Now, let's examine each option:
Option A: In triangle ABC, point E is on AC, and BE is not perpendicular to AC, so BE is not the height.
Option B: In triangle ABC, point B is a right angle, meaning ∠ABC=90°, so AB⊥BC. However, BE is not necessarily perpendicular to AC, and the position of E does not indicate that BE is perpendicular to AC, so it is not the height.
Option C: In triangle ABC, point E is on AC, but there is no indication that BE is perpendicular to AC, so it is not the height.
Option D: In triangle ABC, point E is on AC, and BE is perpendicular to AC (as indicated by the right angle mark at E), so BE is the perpendicular segment from vertex B to side AC, making BE the height of triangle ABC.
Therefore, the correct option is D.
</think>

To determine which figure shows that line segment BE is the height of △ABC, we need to refer to the **definition of the height of a triangle**: the height is the perpendicular segment drawn from a vertex to the opposite side (or its extension), with the foot of the perpendicular lying on the opposite side.

**Analysis of each option:**
- Option A: In △ABC, point E lies on AC, but BE is not perpendicular to AC (there is no right angle mark indicating perpendicularity), so BE is not the height.
- Option B: ∠ABC = 90° (right angle), indicating AB ⊥ BC, but BE is not necessarily perpendicular to AC, and the position of E does not satisfy the condition of "perpendicular to the opposite side," so BE is not the height.
- Option C: There is no indication that BE is perpendicular to AC (no right angle mark), so BE is not the height.
- Option D: Point E lies on AC, and BE ⊥ AC (indicated by the right angle mark at E), meaning BE is the perpendicular segment from vertex B to side AC, satisfying the definition of the height of a triangle.

$$\boxed{D}$$

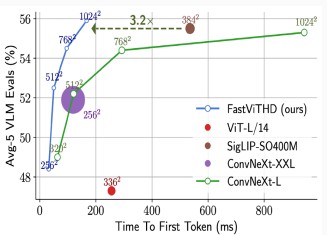

*What does '3.2x' mean in the chart?*

<think>
The user now needs to understand the meaning of "3.2X" in the chart. First, look at the annotation in the chart: "3.2X" is marked next to a point, indicating that the value corresponding to this point is 3.2 times the value of another point. For example, looking at the green line (ConvNext-L), the point corresponding to "3.2X" should be compared with the point on the blue line (FastViT-HD). By observing the values, the point on the blue line is approximately 256², and the point on the green line is approximately 768², so 768/256 = 3, which might be an approximation, but the annotation "3.2X" indicates that the value of this point is 3.2 times the value of another point. Therefore, "3.2X" means that the value corresponding to this point is 3.2 times the value of another point.
</think>

To determine the meaning of "3.2X" in the figure, it is necessary to analyze the annotations in the chart:
1. Locate the annotation "3.2X": In the figure, "3.2X" is marked next to a data point, indicating that the value corresponding to this point is related to another point.
2. Observe the relationship between data points: The green line (ConvNext-L) and the blue line (FastViT-HD) have corresponding data points. By comparing the values of these two lines, it can be found that the value of the green line's data point is approximately 3.2 times the value of the blue line's data point (for example, if the blue line corresponds to a value of 256², the green line corresponds to a value of 768², and (768 ÷ 256 ≈ 3), which may be an approximation, but the annotation "3.2X" clearly indicates that the value of this point is 3.2 times the value of another point).

Therefore, "3.2X" in the figure indicates that the value corresponding to this data point is 3.2 times the value of another data point.

> The value corresponding to this data point is 3.2 times the value of another data point

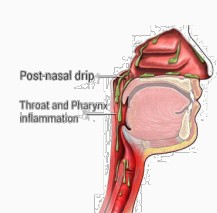

<think>
The user now needs to determine which department to visit for an appointment. First, look at the image, which shows "Throat and Pharynx inflammation," indicating that the issue involves the throat and pharynx. Therefore, it is necessary to find a department related to the throat and pharynx. Typically, in hospitals, ENT (Ear, Nose, and Throat) departments handle such issues. Thus, the corresponding department should be the ENT department.
</think>

*Which hospital department should I go to for an appointment?*

To determine which hospital department to visit for an appointment, we analyze the information in the image. The image highlights "Throat and Pharynx inflammation," which refers to conditions affecting the throat and pharynx. In hospitals, the **ENT (Ear, Nose, and Throat)** department specializes in diagnosing and treating conditions related to these areas.

*ENT (Ear, Nose, and Throat) department*

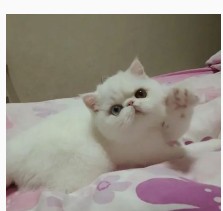

<think>
Okay, let me start by breaking down the question. The user wants a detailed description of the image based on the provided image. The key points mentioned are the white cat with large eyes and fluffy fur, lying on a pink-patterned blanket. It's in a cozy indoor setting. First, I need to focus on the main subject: the cat. The color is white, which is a primary color, so that's straightforward. The mention of "large eyes" and "round pupils" suggests it's looking directly at something or someone, maybe the camera. The fluffy fur adds texture and detail, making the cat look soft. The position on the blanket indicates comfort, perhaps indicating a relaxed environment. The background is simple, neutral tones, which helps the cat stand out. The pink pattern on the blanket adds a touch of color contrast, making the scene more visually appealing. The overall impression is one of warmth and coziness, typical of a pet in a home setting. I should make sure not to include any assumptions beyond what's given. The image doesn't mention any other animals or objects, so I can stick to the cat and the blanket. No need for extra details like the time of day unless specified, but the setting is indoors and neutral, so daylight isn't an issue here. The main elements are the cat, its features, the blanket, and the background. I need to present these in a clear, logical order, starting with the subject (the cat), then its physical characteristics, followed by the setting (blanket and background). Make sure each part flows naturally into the next without being redundant. Avoid any sensory language unless necessary, but since the question is about describing, including colors and textures would be helpful. Double-check that all key points from the original image are covered: white fur, large eyes, fluffy, lying on pink-patterned blanket, indoor, neutral background. Yes, that's all there. Alright, time to put it all together in a coherent description.
</think>

*Describe this image.*

The image depicts a white cat with large, round eyes and fluffy fur, lying comfortably on a bed. The cat's body is oriented diagonally across the frame, with its head raised slightly as if looking directly at the camera. Its front paws are extended outward, giving a sense of curiosity or playfulness. The cat rests on a blanket with a pink floral pattern, which contrasts subtly against the neutral-toned wall in the background. The overall atmosphere is cozy and intimate, capturing a moment of rest in a domestic setting.

| | |
|---|---|
| Direct materials | $15,000 |
| Direct labor | 25,000 |
| Factory depreciation expense | 45,000 |
| Factory utilities expense | 2,000 |
| Payroll staff's salary | 15,000 |

*Extract texts from this image.*

<think>
Okay, let's see. The user wants me to extract the text from the provided image. The image shows a list of costs related to manufacturing. The items are: Direct materials at $15,000, Direct labor $25,000, Factory depreciation expense $45,000, Factory utilities expense $2,000, and Payroll staff's salary $15,000. I need to make sure I copy each line exactly as presented. Let me check each entry again to avoid any mistakes. Direct materials is 15k, then direct labor 25k, followed by factory depreciation 45k, factory utilities 2k, and payroll staff's salary 15k. No commas or extra spaces mentioned in the original text, so just copy them verbatim. Alright, that should be it.
</think>

The extracted text from the image is as follows:

Direct materials - $15,000

Direct labor - $25,000

Factory depreciation expense - $45,000

Factory utilities expense - $2,000

Payroll staff's salary - $15,000

