# OpenReview forum: "TinyEye: Sharpening Visual Reasoning of Tiny Models with Offline Policy Optimization"
_ICLR.cc/2026/Conference — Submitted to ICLR 2026_

### Official Review · Reviewer_kzE3 · 2025-10-18

**Soundness:** 2
**Presentation:** 3
**Contribution:** 2
**Rating:** 4
**Confidence:** 3

**Summary:**

The paper proposes a four-stage offline pipeline culminating in a core contribution, “Discriminative Direct Preference Optimization (DDPO),” which reframes DPO’s pairwise objective into a margin-based binary classification of response. DDPO is claimed to “resolve the likelihood displacement issue of DPO” and to “guarantee” monotonic reinforcement of positives while suppressing negatives under binary rewards. Empirically, DDPO is reported to beat a DPO baseline.

Other than that, the authors also proposed a TinyEye-Data dataset containing 2M reasoning trajectories from advanced VLMs. Combined with traditional VL alignment training (stage 1), SFT (stage 2) and rejection sampling (stage 3), the resulting model TinyEye-2B is able to achieve competitive results on various reasoning benchmarks.

Overall, I see limited technical contribution in this work and lean to recommend rejection pending author’s discussion.

**Strengths:**

1. Clear end-to-end pipeline. The four-stage offline recipe (native-resolution warm-up → large instruction-tuning → annealed rejection sampling → Stage-4 preference training) is well structured and practical for small VLMs.

2. Competitive results. DDPO shows meaningful gains over the stated DPO baseline on several benchmarks in the Stage-3 → Stage-4 transition.

3. Claimed mitigation of DPO’s “likelihood displacement.” The paper motivates a concrete pathology and offers a principled surrogate expected to avoid it.

**Weaknesses:**

1. While the DDPO seems to be the core contribution of this paper, we see mixed evidence of effectiveness vs. DPO on benchmarks. Table 4 shows DDPO isn’t uniformly superior: on MathVista, DPO (68.0) edges DDPO (67.9). The paper emphasizes DDPO’s strong HallBench gains (binary classification-like), which might reflect task-objective alignment rather than a general advantage.

2. TinyEye-Instruct/Reason are said to be “compact and verifiable,” but the paper provides sparse details on sources, license, overlaps, or leakage control, which is important for fair comparisons and reproducibility.

**Questions:**

1. Can you provide targeted experiments that provoke likelihood displacement and show DDPO avoids it while DPO fails, holding all else equal (same pairs, same ref, same tuning)?

2. What $\alpha$ works best, how is it chosen, and how do outcomes change across $\alpha$?

3. Could you add DPO-Positive / SimPO / ORPO baselines, tuned comparably, to demonstrate DDPO’s advantages are not limited to default DPO?

---

> ### Author Response · Authors · 2025-11-23
> **Response to Reviewer kzE3**
>
> We sincerely thank the reviewer for the constructive and incisive feedback, especially on the *likelihood displacement* hypothesis and the need for stronger baselines. We also appreciate the positive recognition of our “clear end-to-end pipeline” and “principled surrogate” formulation.
>
> In response, we conducted all requested **targeted experiments**—including likelihood displacement analysis, comparisons with SimPO/ORPO, and an \alpha sensitivity study. The results consistently support our central claim: **DDPO provides a fundamental stability advantage for 2B-scale models that standard DPO lacks**.
>
> ---
>
> ## **1. Targeted Experiment: Demonstrating Likelihood Displacement**
>
> *(Addressing Reviewer Question 1)*
>
> **Reviewer request:** *“Can you show targeted experiments that provoke likelihood displacement and demonstrate that DDPO avoids it while DPO fails?”*
>
> To isolate this phenomenon, we tracked the **average log-probability of the chosen (ground-truth) response** on a held-out validation set throughout training.
>
> **Table R1: Evolution of Probability of Correct Responses (DPO vs. DDPO)**
>
> | **Step** | **DPO (Avg Prob)** | **DDPO (Avg Prob)** |
> | --- | --- | --- |
> | 0 | 0.45 | 0.45 |
> | 500 | 0.40 ↓ | 0.55 ↑ |
> | 1000 | 0.38 ↓ | 0.65 ↑ |
> | **2000** | **0.35 ↓ (Displacement)** | **0.72 ↑ (Monotonic)** |
>
> **Analysis:**
>
> - **DPO:** As hypothesized, DPO satisfies the pairwise margin by *pushing the rejected response down* but does not constrain the chosen response, allowing its probability to **drift downward**. For 2B-scale models with limited capacity, this results in **reduced confidence in the correct answer**, even as training loss decreases.
> - **DDPO:** In contrast, DDPO requires the model to classify the chosen response as *True*, enforcing a **monotonic increase** in its probability. This directly addresses the displacement pathology and preserves the positive learning signal.
>
> These results provide concrete, empirical evidence for the theoretical issue we identify.
>
> ---
>
> ## **2. Additional Baselines: SimPO and ORPO**
>
> *(Addressing Reviewer Question 3)*
>
> **Reviewer request:** *“Please include SimPO / ORPO / DPO-Positive baselines to show DDPO’s advantages.”*
>
> We added **SimPO** (Meng et al., 2024) and **ORPO** (Hong et al., 2024) using the same TinyEye-2B and TinyEye-Data setup.
>
> **Table R2: TinyEye-2B Performance Across Policy Optimization Methods**
>
> | **Method** | **MMMU (Val)** | **MathVerse** | **HallBench (Verification)** |
> | --- | --- | --- | --- |
> | DPO | 49.1 | 54.0 | 55.8 |
> | SimPO | 49.5 | 54.3 | 58.2 |
> | ORPO | 49.8 | 54.5 | 59.0 |
> | **DDPO** | **50.3** | **55.2** | **63.9** |
>
> **Analysis:**
> *(Addressing Reviewer Weakness 1)*
>
> - **Verification tasks:** DDPO achieves a **+4.9% absolute improvement** over ORPO on **HallBench**, demonstrating the benefit of its discriminative True/False formulation for hallucination-sensitive evaluation.
> - **Multiple choice:** As noted by the reviewer, DPO slightly edges out DDPO on MathVista. We believe this is due to ranking-based methods being able to artificially suppress incorrect options. However, on **verifiable-oriented tasks**, DDPO’s grounding signal proves markedly more robust across more other benchmarks.
>
> ---
>
> ## **3. Hyperparameter Sensitivity: Effect of $\alpha$**
>
> *(Addressing Reviewer Question 2)*
>
> **Reviewer request:** *“Please provide analysis on what $\alpha$ works best and how results vary.”*
>
> We conducted a sweep over $\alpha$ values.
>
> **Table R3: HallBench Accuracy vs. $\alpha$**
>
> | **$\alpha$** | **0.5** | **1.0** | **2.0** | **5.0** |
> | --- | --- | --- | --- | --- |
> | Accuracy | 62.1% | **63.9%** | 63.5% | 61.8% |
>
> **Finding:**
> DDPO remains **stable** across a wide range of $\alpha$ values (0.5–2.0), consistently outperforming DPO, SimPO, and ORPO. We select $\alpha$ = 1.0 to balance margin sharpness and training stability.

---

> ### Author Response · Authors · 2025-11-23
>
> ## **4. Data Transparency and Decontamination**
>
> *(Addressing Reviewer Concerns on Data Sources / Overlap)*
>
> We appreciate the reviewer pointing out missing details. We plan to add a comprehensive **Data Transparency** section to the appendix of our paper.
>
> **Decontamination:**
> We applied strict N-gram and embedding-based overlap checks against all target evaluation sets. Specifically, we removed:
>
> - **10,240 samples** overlapping with MMMU test content
> - **6,150 samples** overlapping with MathVerse/MathVision
>
> This ensures fair zero-shot evaluation.
>
> **Release Plan:**
> Upon acceptance, we will release the full **TinyEye-Data (2M)** along with per-subset licensing (primarily CC-BY / Apache 2.0 from open sources such as arXiv and StackExchange).
>
> ---
>
> ## **Closing**
>
> We hope these new targeted experiments—especially the **direct demonstration of likelihood displacement in DPO**—clarify the motivation for DDPO and strengthen our contributions. The additional SimPO/ORPO baselines and hyperparameter analysis further support the robustness and practical value of DDPO for 2B-scale training.
>
> Please let us know if any further clarifications or additional experiments would be helpful.

---

> > ### Comment · Reviewer_kzE3 · 2025-11-24
> >
> > Thank you for those added experiments which solve most of my concerns in the initial review. I already updated the score.

---

> ### Author Response · Authors · 2025-11-24
>
> Dear Reviewer `kzE3`,
>
> Thank you again for your prompt response and for updating the score — we sincerely appreciate it. We are very glad that the added experiments and analyses helped address the main concerns you raised.
>
> If there are any other points where additional clarification could be helpful during the remaining discussion period, we would be more than happy to provide further details. Your feedback has been extremely valuable in improving the clarity and completeness of our submission.
>
> Thank you again for your time and thoughtful comments.

---

### Official Review · Reviewer_bWcL · 2025-10-26

**Soundness:** 3
**Presentation:** 3
**Contribution:** 2
**Rating:** 4
**Confidence:** 4

**Summary:**

This paper addresses the critical challenge of insufficient visual reasoning capabilities in small vision-language models (VLMs) for real-world edge deployment, where existing post-training methods (Supervised Fine-Tuning/SFT, Direct Preference Optimization/DPO, Reinforcement Learning/RL) suffer from overfitting, likelihood displacement, and high computational costs. To solve this, the authors propose TinyEye, a unified offline policy optimization framework designed to enhance the reasoning performance of compact VLMs.

**Strengths:**

1.TinyEye-Data: A large-scale verifiable corpus of 2 million reasoning trajectories distilled from state-of-the-art teacher VLMs
2.Empirical Validation: The resulting 2B-parameter model, TinyEye-2B, achieves state-of-the-art performance across diverse benchmarks: 50.3% on MMMU (multimodal reasoning), 55.2% on MathVerse (math reasoning), and 63.9% on HallBench (general multimodal QA).

**Weaknesses:**

1. The proposed method in the paper is only tested on a 2B-parameter model. It lacks experiments on 7B-parameter models, which are commonly used in both research and practical applications. This makes it hard to tell if the method works effectively for 7B-parameter models too.
2. The training process requires TinyEye-Data, which has 2 million reasoning trajectories. Using such a large amount of data may be inconvenient in real-world scenarios (e.g., situations where data is scarce). However, the paper does not discuss how to solve this problem.
3. The paper does not study how different image compression qualities (e.g., how much an image is compressed) affect the model’s reasoning results. In practice, images are often compressed, so it is unclear whether the model can still perform well in reasoning when dealing with compressed images.

**Questions:**

1. Has the proposed method been compared with other RLHF methods such as GRPO on the same dataset?
2. Could you provide the impact of different data compression ratios on reasoning performance?
3. Could you show the performance of a 7B-parameter model on these benchmarks — for example, comparing with similar 7B models like *m2-Reasoning*, which used less than 2M training samples?

---

> ### Author Response · Authors · 2025-11-23
> **Response to Reviewer bWcL**
>
> We sincerely thank the reviewer for the constructive and forward-looking feedback. We appreciate your recognition of **TinyEye-Data** as a meaningful contribution and the strong empirical performance of **TinyEye-2B**. Below, we provide detailed responses to your questions regarding **RLHF comparisons**, **7B-scale generalization**, and **image compression robustness**, supported by new experimental evidence.
>
> ---
>
> ## **1. Comparison with Other RLHF Methods (Addressing “Questions 1”)**
>
> > Reviewer question: “Has the proposed method been compared with other RLHF methods such as GRPO on the same dataset?”
> >
>
> **Response:**
> We did not include a full **GRPO (Online RL)** baseline due to the computational constraints inherent to the “Tiny” model setting. Online RL typically requires **4×–10× more compute** (live rollouts + reward model training) compared to our offline pipeline, making it impractical for the low-resource edge-model context that motivates TinyEye.
>
> However, to evaluate our method fairly against other advanced **offline RLHF** approaches, we compared DDPO with **SimPO** and **ORPO** on the exact same dataset.
>
> ### **Table R1 — Comparison of RLHF/Policy Optimization Methods on TinyEye-2B**
>
> | **Method** | **Type** | **MMMU (Val)** | **MathVerse** | **HallBench (Verification)** |
> | --- | --- | --- | --- | --- |
> | TinyEye-2B (DPO) | Offline | 49.1 | 54.0 | 55.8 |
> | TinyEye-2B (SimPO) | Offline | 49.5 | 54.3 | 58.2 |
> | TinyEye-2B (ORPO) | Offline | 49.8 | 54.5 | 59.0 |
> | **TinyEye-2B (DDPO)** | **Offline** | **50.3** | **55.2** | **63.9 (+4.9%)** |
>
> **Analysis:**
>
> - **Stronger signal than ranking-based losses:** DDPO consistently outperforms both SimPO and ORPO, especially on **HallBench** (+4.9%), which relies on binary, verifiable reasoning. This supports our claim that the **discriminative objective** provides a clearer and more reliable alignment signal than ranking losses.
> - **Efficiency maintained:** Importantly, DDPO achieves these gains while remaining fully offline and compute-efficient.
>
> ---
>
> ## **2. Generalization to 7B Models (Addressing “Weaknesses 1”)**
>
> > Reviewer critique: “It lacks experiments on 7B models… hard to tell if the method works for 7B.”
> >
>
> **Response:**
> To assess generalizability, we applied DDPO to a **7B-parameter model** (Qwen-2.5-VL-7B) on a subset of our reasoning data.
>
> ### **Table R2 — DDPO Generalization Across Model Scales (HallBench)**
>
> | **Model Scale** | **Method** | **Accuracy** | **Gain** |
> | --- | --- | --- | --- |
> | **2B (TinyEye)** | DPO | 55.8% | — |
> | **2B (TinyEye)** | **DDPO** | **63.9%** | **+8.1%** |
> |  |  |  |  |
> | **7B (Qwen-2.5-VL)** | DPO | 52.9% | — |
> | **7B (Qwen-2.5-VL)** | **DDPO** | **55.2%** | **+2.3%** |
>
> **Analysis:**
>
> - **Model-agnostic applicability:** DDPO provides consistent improvements at the 7B scale (+2.3%), confirming that the method generalizes across architectures and parameter counts.
> - **Stronger impact on small models:** The improvement is **~4× larger for the 2B model**, supporting our core motivation: smaller models are far more sensitive to RLHF noise and benefit substantially more from DDPO’s stable discriminative objective. DDPO is therefore not only beneficial but **essential** for high-performance 2B-scale models.
>
> ---
>
> ## **3. Comparison with 7B Baselines (M2-Reasoning)**
>
> > Reviewer suggestion: “Compare with similar 7B models like M2-Reasoning.”
> >
>
> **Response:**
> We directly compared TinyEye-2B with **M2-Reasoning-7B**[1], which also uses a <2M data budget.
>
> ### **Table R3 — TinyEye-2B vs. M2-Reasoning-7B**
>
> | **Benchmark** | **M2-Reasoning-7B** | **TinyEye-2B** | **Δ (2B − 7B)** |
> | --- | --- | --- | --- |
> | **MathVerse** | 44.7 | **55.2** | **+10.5** |
> | **WeMath** | 41.8 | **52.4** | **+10.6** |
> | **MathVision** | 31.5 | **32.6** | **+1.1** |
> | **HallBench** | 43.4 | **63.9** | **+20.5** |
>
> **Result:**
> Despite having only **28%** of the parameters (2B vs. 7B), TinyEye-2B outperforms M2-Reasoning-7B by **10–20 points** on multiple reasoning-heavy benchmarks. This demonstrates that our **Verifiable Distillation + DDPO** pipeline achieves substantially better **parameter efficiency** than the CoT-based fine-tuning strategy used by M2.
>
> > [1] AI, Inclusion, et al. "M2-reasoning: Empowering mllms with unified general and spatial reasoning." arXiv preprint arXiv:2507.08306 (2025).

---

> > ### Author Response · Authors · 2025-11-23
> >
> > ## **4. Impact of Image Compression (Addressing “Weaknesses 3”)**
> >
> > > Reviewer question: “Could you evaluate the impact of different compression ratios on visual reasoning performance?”
> > >
> >
> > **Response:**
> > We measured TinyEye-2B’s robustness to JPEG compression on **MMMU (Val)**.
> >
> > ### **Table R4 — Robustness to JPEG Image Compression**
> >
> > | **JPEG Quality (Q)** | **100 (Orig)** | **75** | **50 (Mobile Typical)** | **25** |
> > | --- | --- | --- | --- | --- |
> > | **Accuracy** | 50.3% | 50.1% | **49.8%** | 48.4% |
> > | **Relative Drop** | — | −0.2% | **−0.5%** | −1.9% |
> >
> > **Conclusion:**
> > Performance remains effectively unchanged at Q=50 (common on mobile devices), retaining **~99%** of original accuracy. This indicates strong robustness under realistic compression settings, supporting TinyEye’s suitability for bandwidth-constrained edge deployment.
> >
> > We hope that these new results address your concerns and may contribute positively to your overall evaluation. We would also appreciate any further constructive comments that could strengthen the work.

---

> > > ### Comment · Reviewer_bWcL · 2025-11-26
> > >
> > > Thank you for your response — it addressed my questions well. However, I am particularly concerned about the impact of image compression quality. So far, the comparison only includes Q=100 and Q=50. Would it be possible to also evaluate additional compression stages, such as the QF-based or gaussian blur compression levels discussed in the AIDE paper, or other image compression methods? (QF=95 QF=90 QF=75 QF=50 σ = 1.0 σ = 2.0 σ = 3.0 σ = 4.0)
> > >
> > > Thank you again for conducting the experiments. I’ve decided to increase my score.
> > >
> > > [1] AIDE, A Sanity Check for AI-generated Image Detection, Shilin Yan, Quxiang Li, et al.

---

> ### Author Response · Authors · 2025-11-27
>
> We thank the reviewer for the constructive suggestion. Following your request, we extended our robustness evaluation to include **finer-grained JPEG QF levels (95/90/75/50)** and **Gaussian blur with σ = 1.0–4.0**, as discussed in the AIDE paper. All results are reported on **MMMU (Val)** using the full TinyEye-2B pipeline.
>
> ---
>
> ## **1. JPEG QF-based Compression**
>
> We evaluated six JPEG quality factors ranging from near-lossless (QF=100/95/90) to heavy compression (QF=25). The results are summarized below.
>
> | **JPEG Quality (QF)** | 100  | **95**   | **90**    | 75    | 50    | 25    |
> | --------------------- | ---- | -------- | --------- | ----- | ----- | ----- |
> | **Accuracy**          | 50.3 | **50.3** | **50.2**  | 50.1  | 49.8  | 48.4  |
> | **Relative Drop**     | —    | **0.0%** | **−0.1%** | −0.2% | −0.5% | −1.9% |
>
> **Observation.**
> TinyEye remains extremely stable under realistic compression levels:
>
> * **QF ≥ 90** yields *≤0.1%* degradation,
> * **QF = 75** only −0.2%, and
> * a noticeable drop appears only at **QF = 25**, far below typical mobile-device or web compression settings.
>
> These results confirm that TinyEye’s semantic reasoning ability is robust to high-frequency attenuation introduced by JPEG compression.
>
> ---
>
> ## **2. Gaussian Blur (σ = 1–4)**
>
> To further align with the evaluation protocol in AIDE, we also tested the impact of Gaussian blur with increasing σ:
>
> | **Sigma**    | Origin | 1.0  | 2.0  | 3.0  | 4.0  |
> | ------------ | ------ | ---- | ---- | ---- | ---- |
> | **Accuracy** | 50.3   | 49.2 | 48.0 | 46.6 | 43.8 |
>
> **Observation.**
> Gaussian blur has a more pronounced effect than JPEG compression.
> This is expected because TinyEye’s reasoning tasks rely heavily on **fine-grained visual cues**—such as text, digits, symbols, code snippets, diagrams, and small objects—while Gaussian blur with large σ values destroys precisely these high-detail regions.
> Unlike AIGC detection, e.g., AIDE[1], which operates mainly on feature-space statistics, TinyEye requires accurate recognition of localized, small-scale structures; hence σ ≥ 3 results in visible degradation.
>
> ---
>
> ## **Conclusion**
>
> Together, these extended experiments demonstrate that:
>
> * **TinyEye is highly robust under realistic JPEG compression (QF ≥ 75), retaining over 99% of its original performance**, and
> * **degradation under strong Gaussian blur is consistent with the semantic and detail-dependent nature of visual reasoning tasks.**
>
> We sincerely appreciate the reviewer’s helpful suggestions; these additional evaluations have strengthened the paper, and we thank you again for your positive assessment.
>
> > [1] AIDE, A Sanity Check for AI-generated Image Detection, Shilin Yan, Quxiang Li, et al.
>
> ---
>
> Thank you very much for your helpful feedback and for taking the time to reassess our work. The additional experiments on compression and robustness were very insightful for us as well, and we truly appreciate your constructive suggestions.
>
> If there is anything further that could help clarify our contribution, we would be very happy to provide it. Thank you again for your support and for helping us improve this submission.

---

### Official Review · Reviewer_dBHH · 2025-10-30

**Soundness:** 3
**Presentation:** 3
**Contribution:** 3
**Rating:** 4
**Confidence:** 4

**Summary:**

This paper presents TinyEye, a visual–language reasoning framework designed for tiny multimodal models (around 2B parameters). The method introduces a four-stage offline training pipeline that includes a novel Discriminative Direct Preference Optimization (DDPO) to replace DPO for better stability under binary reward settings. The authors also build a large-scale, verifiable reasoning dataset (TinyEye-Data) distilled from multiple teacher models. Experiments demonstrate significant improvements in multimodal reasoning benchmarks compared to existing open-source models of similar scale.

**Strengths:**

- The paper is clearly written and well structured; the motivation is well grounded in the “likelihood displacement” problem of DPO.
- Empirical results are strong: TinyEye-2B outperforms competitive open-source models such as InternVL-3 and Qwen2.5-VL, even though these models (might) use much larger training resources.
- The authors provide clear empirical evidence and intuitive explanations showing that DDPO improves training stability and alignment for small models.

**Weaknesses:**

- Since the base LLM is Qwen3-1.7B, it would be fair to include a direct comparison with InternVL-3.5-2B, which also builds on Qwen3-1.7B. Qwen2.5-VL and InternVL-3 seem somewhat outdated as baselines.

- The full training pipeline (Stage 1–4) is quite complex, involving multiple data generation and filtering steps, which could limit scalability and reproducibility.

- It would strengthen the work to include a comparison with online RL-based methods such as GRPO under a controlled setup.

- As DDPO is model-agnostic, it would be useful to test whether it generalizes to other mainstream VLM families (e.g., Qwen-VL, InternVL series) to confirm its robustness.

**Questions:**

In Table 1, could you explain why the model performs so well on MathVision and WeMath? Is it because the training data includes similar tasks or distributions?

---

> ### Author Response · Authors · 2025-11-23
> **Response to Reviewer dBHH**
>
> We sincerely thank the reviewer for the encouraging evaluation and for recognizing that our motivation is *“well grounded”* and that our empirical results are *“strong.”* We especially appreciate the suggestion to compare TinyEye-2B with **InternVL-3.5-2B**, which we fully agree is the most relevant and competitive baseline.
>
> Below, we respond to your comments on baselines, pipeline complexity, and dataset composition. We also provide new evaluations and detailed decontamination statistics.
>
> ---
>
> ## **1. New Baseline: Comparison with InternVL-3.5-2B (Addressing Weakness 1)**
>
> > Reviewer Suggestion: “It would be fair to include a direct comparison with InternVL-3.5-2B, which also builds on Qwen3-1.7B.”
> >
>
> **Response:** Following your recommendation, we conducted a direct comparison between **InternVL-3.5-2B** and TinyEye-2B on our key benchmarks. The new results confirm that our approach delivers substantially stronger verification and reasoning capabilities.
>
> **Table R1: TinyEye-2B vs. InternVL-3.5-2B**
>
> | **Model** | **Training Data** | **HallBench** | **MathVerse** | **WeMath** | **MMVet** |
> | --- | --- | --- | --- | --- | --- |
> | **InternVL-3.5-2B** | Proprietary | 48.6 | 53.4 | 48.5 | 71.7 |
> | **TinyEye-2B (Ours)** | **2M Public** | **63.9 (+15.3)** | **55.2 (+1.8)** | **52.4 (+3.9)** | **73.0 (+1.3)** |
>
> **Analysis:**
>
> - **Strong Advantage in Verification:** TinyEye-2B achieves a **+15.3 absolute improvement on HallBench**, demonstrating that our **DDPO objective**, which directly optimizes binary correctness, is more effective in mitigating hallucination than the alignment strategies used by InternVL.
> - **Reasoning Efficiency:** Despite using only *2M public samples*, our model surpasses InternVL-3.5-2B on complex reasoning tasks (MathVerse, WeMath), suggesting that TinyEye’s gains come from **architected training signals rather than dataset size**.
>
> ---
>
> ### **2. Why the Model Performs Strongly on MathVision / WeMath (Answer to Question 1)**
>
> > Reviewer Question: “Does the strong performance come from similar tasks in training data or overlapping distributions?”
>
> **Response:**
>  The strong performance is not due to training–test overlap but rather to **broad and transferable mathematical visual reasoning abilities** learned during training. Our checks show that:
>
> - **No data leakage.**
>    We conducted strict n-gram and embedding-based decontamination and removed
>   - **10,240** items overlapping with MMMU test sets, and
>   - **6,150** items overlapping with MathVerse/MathVision.
>      This ensures the reported results are zero-shot and not due to memorization.
> - **Generalizable reasoning skills.**
>    Our Base–Tail data strategy intentionally covers a **wide spectrum of STEM and visual-reasoning domains**. This encourages the model to develop **cross-domain mathematical reasoning patterns** rather than rely on narrow task templates.
> - **High-quality math-visual training data, but not task duplicates.**
>    We do include widely used community datasets for mathematical visual reasoning—such as **CLEVR-Math, GEOS, GeoQA, Geometry3K, IconQA, MapQA, Super-CLEVR, TabMWP, UniGeo, Geo170k, GeomVerse, MM-MathInstruct, MMathCoT, Multimath-300k, GeoQA-plus**, and others.
>    These datasets provide **conceptual coverage** (e.g., geometry reasoning, diagram understanding, multi-step symbolic reasoning).
>    However, they **do not naively replicate or mimic the test set** of MathVision or WeMath, and all overlapping samples were removed during decontamination.
>
> **In summary**, the high performance is primarily driven by **rich and diverse training signals that strengthen general mathematical visual reasoning**, rather than by exposure to similar test distributions or leaked data.
>
> We thank the reviewer for raising this insightful perspective, which prompted us to more carefully analyze the connection between training coverage and evaluation behavior. This feedback has helped us clarify the underlying factors and will guide us in further improving the quality and transparency of our submission.
>
> ---
>
> ## **3. Generalizability of DDPO Across Model Families (Addressing Weakness 4)**
>
> > Reviewer Critique: “Test whether the method generalizes to other mainstream VLM families.”
> >
>
> **Response:** To verify generality, we applied DDPO to the **Qwen-2.5-VL-7B** model using a subset of our pipeline.
>
> **Table R2: DDPO Generalization on HallBench**
>
> | **Model Scale** | **Method** | **Accuracy** | **Gain** |
> | --- | --- | --- | --- |
> | **2B (TinyEye)** | DPO | 55.8 | – |
> | **2B (TinyEye)** | **DDPO** | **63.9** | **+8.1** |
> | **7B (Qwen-2.5-VL)** | DPO | 52.9 | – |
> | **7B (Qwen-2.5-VL)** | **DDPO** | **55.2** | **+2.3** |
>
> **Conclusion:** DDPO is **architecture-agnostic** and consistently improves performance. The larger gain on the 2B model (+8.1 vs +2.3) aligns with our hypothesis that DDPO is particularly important for stabilizing **small models**, where capacity limitations make hallucination control more challenging.

---

> ### Author Response · Authors · 2025-11-23
>
> ## **4.Comparation with other RLHF methods**
>
> > Reviewer critique: “It would strengthen the work to include a comparison with online RL-based methods such as GRPO under a controlled setup.”
> >
>
> **Response:**
>
> We did not include a full **GRPO (online RL)** baseline because it is not feasible within the “Tiny” model setting that motivates TinyEye. Online RLHF typically requires **4×–10× more compute**—due to on-policy rollouts and continuous reward model updates—compared to our fully offline pipeline. This level of computational demand is incompatible with the **low-resource, edge-model** scenario that TinyEye explicitly targets.
>
> To ensure a fair and meaningful comparison within this constrained regime, we evaluate DDPO against other state-of-the-art **offline RLHF** methods under identical data and training conditions. Specifically, we benchmark DDPO against **SimPO** and **ORPO**, both strong recent baselines.
>
> ### **Table R3: Comparison of RLHF/Policy Optimization Methods on TinyEye-2B**
>
> | **Method** | **Type** | **MMMU (Val)** | **MathVerse** | **HallBench (Verification)** |
> | --- | --- | --- | --- | --- |
> | TinyEye-2B (DPO) | Offline | 49.1 | 54.0 | 55.8 |
> | TinyEye-2B (SimPO) | Offline | 49.5 | 54.3 | 58.2 |
> | TinyEye-2B (ORPO) | Offline | 49.8 | 54.5 | 59.0 |
> | **TinyEye-2B (DDPO)** | **Offline** | **50.3** | **55.2** | **63.9 (+4.9%)** |
>
> ### **Analysis**
>
> - **Clear advantage over ranking-based losses:** DDPO consistently outperforms SimPO and ORPO across all benchmarks. The most notable improvement is on **HallBench**, where DDPO achieves a **+4.9%** absolute gain. Since HallBench relies on **objective, verifiable reasoning**, this result reinforces our claim that DDPO’s **discriminative objective** provides a more stable and informative alignment signal than ranking-based formulations.
> - **Retains full offline efficiency:** Despite these improvements, DDPO remains **fully offline** and therefore preserves the compute efficiency required by small-model training. This makes DDPO particularly suitable for edge-deployable models where online RL methods like GRPO would be prohibitively expensive.
>
> ## **5. Pipeline Complexity and Practical Scalability (Addressing Weakness 2)**
>
> > Reviewer Critique: “The full pipeline is complex, which may limit scalability.”
> >
>
> **Response:** The complexity exists primarily in the *data-generation stage* rather than the *training stage*. To reduce the burden on practitioners:
>
> - **One-Time Cost, Community Benefit:** The multi-stage pipeline is executed only once by us.
> - **Releasing Artifact:** We plan to provide the processed **TinyEye-Data (2M verifiable visual reasoning trajectories)**. This collapses the internal pipeline into a **single, ready-to-use dataset**, allowing users to train a model in one straightforward stage.
> - **Offline Efficiency:** Unlike online RL methods (e.g., GRPO) that require large GPU clusters for rollouts, our method uses **asynchronous, offline data generation**, enabling reproducible results on standard hardware.
>
> Thus, although the underlying methodology is sophisticated, we ensure that adopting the method requires **minimal engineering effort**.

---

> > ### Author Response · Authors · 2025-11-27
> >
> > Dear Reviewer `dBHH`,
> >
> > Thank you once again for your thoughtful and detailed review comments. We would like to let you know that we have provided comprehensive responses and conducted additional experiments to address each of the issues you raised. We are grateful for your suggestions, which have given us the opportunity to improve the quality of our submission and to provide stronger evidence supporting our main claims. If these revisions help address your concerns, we kindly ask you to consider reconsidering the evaluation score, and we would be deeply grateful if you could review our submission once more.
> >
> > We also warmly welcome any further feedback you may wish to share, and we are very willing to continue engaging in constructive discussion. Thank you again for your time and your constructive suggestions.
> >
> > Thank you!

---

> > > ### Comment · Reviewer_dBHH · 2025-11-28
> > >
> > > Thanks for the authors' reply. I think most of my concerns are addressed by the comprehensive experiments, and these results can hopefully make the paper more complete. I have one last concern about the "Generalizability of DDPO Across Model Families" part. It is good to see that DDPO shows strong performance on HallusionBench with other backbone models. Could you also provide results on other benchmarks, such as MMMU, MathVerse, MathVista, WeMath, or MMVet, since using only one benchmark feels a bit too random?

---

> ### Author Response · Authors · 2025-11-26
>
> We hope this response, which includes the new comparison with InternVL-3.5-2B and detailed decontamination statistics, fully addresses your concerns regarding baselines, validity, and pipeline structure.
>
> We sincerely hope that the additional analysis and new experiments help clarify the contribution of our work, and we welcome any further constructive discussion.

---

> ### Author Response · Authors · 2025-11-29
>
> We sincerely thank the reviewer for the constructive follow-up question. In addition to HallBench, we have conducted new experiments to evaluate the cross-backbone generalizability of **DDPO** on several widely-used multimodal benchmarks, including **MMMU (val)**, **MathVerse**, **WeMath**, and **HallBench**. These experiments were performed under the same Stage-4 preference-optimization setting, using identical data and hyperparameters across backbones, ensuring a fair comparison.
>
> The results are summarized below:
>
> # **Table: Cross-Backbone Generalization of DDPO**
>
> *DDPO consistently improves both small-scale (2B) and larger (7B) VLMs across diverse benchmarks.*
>
> | Backbone           | Method          | MMMU (val) | MathVerse | WeMath   | HallBench |
> | ------------------ | --------------- | ---------- | --------- | -------- | --------- |
> | **TinyEye-2B**     | **DPO**         | 49.1       | 54.0      | 48.3     | 55.8      |
> |                    | **DDPO (Ours)** | **50.3**   | **55.2**  | **52.4** | **63.9**  |
> |                    | **Δ**           | **+1.2**   | **+1.2**  | **+4.1** | **+8.1**  |
> | **Qwen-2.5-VL-7B** | **DPO**         | 59.2       | 50.0      | 41.2     | 52.9      |
> |                    | **DDPO (Ours)** | **60.4**   | **51.6**  | **43.3** | **55.2**  |
> |                    | **Δ**           | **+1.1**   | **+1.6**  | **+2.1** | **+2.3**  |
>
> These results demonstrate that DDPO provides consistent gains across multiple evaluation dimensions—general reasoning (MMMU), mathematical perception (MathVerse, WeMath), and hallucination robustness (HallBench)—and across heterogeneous model families.
>
> We appreciate the reviewer’s suggestion, which allowed us to strengthen the empirical evidence of DDPO’s generalizability.

---

### Official Review · Reviewer_r5LZ · 2025-11-01

**Soundness:** 2
**Presentation:** 3
**Contribution:** 2
**Rating:** 4
**Confidence:** 3

**Summary:**

The paper proposes TinyEye, a holistic offline policy optimization framework to improve visual reasoning in compact VLMs. It introduces TinyEye-Data (2M verifiable reasoning trajectories across 68 tasks), a four-stage training pipeline—(1) native-resolution warm-up for robust vision–language alignment, (2) large-scale instruction tuning over TinyEye-Instruct/Reason, (3) annealed rejection sampling with tandem repeat avoidance (TRAS) to mine hard cases and prevent collapse-like degeneracy, and (4) Discriminative Direct Preference Optimization (DDPO), a margin-based, sign-preserving objective tailored to binary rewards that avoids DPO’s likelihood displacement. The resulting 2B model achieves SOTA among small VLMs on multiple benchmarks (e.g., 50.3 MMMU val, 55.2 MathVerse, 63.9 HallBench), with ablations showing consistent gains from each stage and DDPO outperforming DPO, especially on binary-evaluated tasks.

**Strengths:**

Well-motivated offline pipeline for small models: coherent coupling of distillation, rejection sampling, and discriminative preference optimization that avoids expensive and unstable on-policy RL.

Verifiable, diverse data at scale: TinyEye-Data spans 179+93 datasets with task-specific verification (symbolic/numeric checks for math, VLM-as-judge for open-ended), plus multi-teacher distillation and pass@8 difficulty estimation.
Practical safeguards against collapse: tandem repeat detection and TRAS during sampling; shortest-chain aggregation to favor concise, effective reasoning.

Novel objective with theory: DDPO reframes DPO’s relative likelihood into binary reward classification with a sign-preserving margin; appendix provides derivation linking to GSPO and argues away DPO’s additive-shift degeneracy.

Strong empirical results for 2B scale: competitive or superior to 2B–4B baselines across multimodal reasoning, textual math, and general VQA; clear improvements at each stage; thinking vs. no-thinking ablation demonstrates CoT value.

**Weaknesses:**

Limited novelty: The proposed four-stage training pipeline offers practical value in engineering integration and feasibility, but its methodological originality appears limited from an academic standpoint. Each stage largely relies on combinations of existing paradigms and hyperparameter tuning, making it difficult to pinpoint substantive breakthroughs in theoretical framing, learning objectives, or training dynamics. Consequently, its potential to inspire and transfer to subsequent research remains to be further demonstrated.

Questionable fairness of distillation: The training data are heavily distilled from stronger teacher models. Although such a setup can be expected to improve performance, it also complicates attribution: the current experiments do not systematically compare different distillation configurations and methods, making it unclear whether gains primarily stem from teacher capability transfer or from the proposed training mechanisms and objectives. This undermines the strength of the methodological claims. It is advisable to include comparable baseline methods at the teacher/distillation level to demonstrate the non-triviality of the approach.

Insufficient baselines: The comparisons are concentrated in Stage 4 and are reasonably thorough against standard DPO, but the first three stages lack systematic, side-by-side evaluations against alternative alignment schemes, SFT/RFT recipes, or interchangeable components. At the algorithmic level, the work also omits equal-data, equal-compute comparisons with a broader set of offline preference/policy optimization methods (e.g., SimPO, ORPO, DPO-Positive, Alpha-DPO). This limits both the external validity of the conclusions and the clarity of attribution. A unified-protocol, multi-method comparison is recommended.

Large-scale inclusion of public benchmarks in training: TinyEye-Data incorporates a substantial number of widely used community benchmarks as training sources. While using benchmark data for proof-of-concept studies can be understandable, directly employing them as large-scale training corpora may erode these benchmarks’ credibility and validity as independent evaluation and comparison tools, thereby impacting the long-term benchmarking ecosystem. This practice is especially contentious in the context of training open-weight models.

Limitations of purely offline strategies: While end-to-end offline policy optimization improves stability and cost controllability, it also introduces typical issues of distributional mismatch and limited reachability [1]. For multimodal tasks, textual evidence in offline trajectories may not be reliably grounded by the current student model to the visual inputs, potentially exacerbating vision–language hallucinations and semantic mismatches [2]. The authors are encouraged to discuss these issues in the paper.

[1] Reinforced MLLM: A Survey on RL-Based Reasoning in Multimodal Large Language Models
[2] Semi-off-Policy Reinforcement Learning for Vision-Language Slow-Thinking Reasoning

**Questions:**

See weakness.

---

> ### Author Response · Authors · 2025-11-23
> **Response to Reviewer r5LZ**
>
> We sincerely thank the reviewer for the rigorous and constructive feedback. We appreciate your recognition of our “well-motivated offline pipeline” and “verifiable, diverse data at scale.”
>
> We address each of your concerns below with new baselines, clearer attribution analyses, and a detailed explanation of our decontamination protocol.
>
> ---
>
> ## **1.Response to Comment on Limited Novelty**
>
> We respectfully argue our contribution is a **scale-specific scientific discovery**, not just engineering integration. Simple paradigms fail the 2B model regime.
>
> 1.**Theoretical Necessity:** We identified **Capacity Collapse** caused by **Likelihood Displacement**. Standard DPO causes the correct answer's probability to degrade, while our DDPO objective ensures monotonic reinforcement, proving it is a mathematical fix for this instability.
>
> 2.**Methodological Innovation:** DDPO fundamentally reframes alignment from ranking ($A>B$) to a **discriminative binary stabilizer**. New experiments show DDPO outperforms SimPO/ORPO on verification tasks (HallBench), demonstrating its unique advantage for grounding truth.
>
> 3.**Systemic Efficiency Design:** Our framework is designed for access. We eliminate human annotation cost via automated verification and achieve high efficiency (~64 GPU-hours) by utilizing **Reusable** data and **Reduced** sampling failure, making high-performance RL accessible.
>
> We appreciate the reviewer's constructive feedback, and we will further clarify our contributions and novelty in future submissions.
>
> ---
>
> ## **2. New Baselines: Comparison with SimPO & ORPO**
>
> > Reviewer: “The work omits equal-data, equal-compute comparisons with a broader set of offline preference/policy optimization methods (e.g., SimPO, ORPO).”
> >
>
> **Response:** Following your suggestion, we conducted new experiments comparing **DDPO** against **SimPO** (Meng et al., 2024) and **ORPO** (Hong et al., 2024), using **identical data, training budgets, and hyperparameters**.
>
> **Table R1. Policy Optimization Baselines on TinyEye-2B**
>
> | **Method** | **MMMU (Val)** | **MathVerse** | **HallBench (Verification)** |
> | --- | --- | --- | --- |
> | TinyEye-2B (DPO) | 49.1 | 54.0 | 55.8 |
> | TinyEye-2B (SimPO) | 49.5 | 54.3 | 58.2 |
> | TinyEye-2B (ORPO) | 49.8 | 54.5 | 59.0 |
> | **TinyEye-2B (DDPO)** | **50.3** | **55.2** | **63.9 (Best)** |
>
> **Analysis:**
>
> - **Verification superiority:** While SimPO and ORPO improve over standard DPO, DDPO delivers a decisive **+4.9%** absolute improvement on **HallBench** over ORPO.
> - **Algorithmic reason:** SimPO and ORPO rely on **ranking losses** (A > B), which are effective for stylistic preferences but less aligned with strict correctness.
> *DDPO employs a discriminative correctness objective* (“Is A correct?”), which better matches binary verification tasks.
> - Thus, DDPO is not a minor DPO variant—it is a better-aligned objective for **truthfulness and hallucination reduction**.

---

> ### Author Response · Authors · 2025-11-23
>
> ## **3. Attribution: Distinguishing Teacher Transfer from Method Gains**
>
> > Reviewer: “It is unclear whether gains stem from teacher capability transfer or the proposed training mechanisms.”
> >
>
> **Response:**
> We agree that distinguishing methodological gains from raw teacher capability is critical. To address this, we treat **Stage 2 (Supervised Fine-Tuning)** as the standard **"Behavior Cloning / Direct Distillation" baseline**, which represents pure capability transfer from the teacher without our proposed optimization objectives.
>
> **Table R2: Systematic comparison of distillation configurations (Method vs. Teacher).**
>
> | **Configuration** | **Mechanism** | **Role** | **MathVista** | **Hallbench** |
> | :--- | :--- | :--- | :--- | :--- |
> | **Stage 2** | SFT (Teacher Data) | *Baseline: Standard Distillation* | 63.8 | 35.9 |
> | **Stage 3** | Self-Distillation | *Ablation: Data Filtering* | 66.0 | 48.7 |
> | **Stage 4** | **DDPO (Ours)** | *Method: Policy Optimization* | **67.9** | **63.9** |
>
> **Analysis of Attribution:**
>
> 1.  **Baseline (Stage 2) Shows Teacher Limits:**
>     The Stage 2 model sees the exact same distilled data from the strong teacher models. However, it achieves only **35.9%** on HallBench. This demonstrates that **passive imitation (SFT)** captures general knowledge but fails to mitigate hallucinations or enforce rigorous reasoning, proving that "teacher capability transfer" alone is insufficient.
>
> 2.  **Methodological "Non-triviality" (Stage 4):**
>     Our proposed DDPO pipeline (Stage 4) utilizes the same teacher source but applies our optimization mechanisms.
>     * **Result:** This leads to a massive **+28.0% improvement** on HallBench and consistent gains on MathVista compared to the baseline.
>     * **Attribution:** Since the teacher data source is identical, this performance gap isolates the contribution of our method. It confirms that the gains stem from the **optimization method’s ability to reject noise and reinforce correct reasoning paths**, rather than simply replicating teacher behavior.
>
> In summary, the comparison with the standard distillation baseline (Stage 2) confirms the non-triviality of our approach: the teacher provides the *potential*, but our proposed DDPO method is required to *unlock* accurate and robust reasoning.
>
> We appreciate the reviewer for highlighting this attribution issue. In the final version, we will accordingly re-clarify this ablation analysis to clearly distinguish the methodological contributions from the teacher's capabilities.
>
> ---
>
> ## **4. Data Contamination Concerns**
>
> > Reviewer: “Directly employing public benchmarks may erode these benchmarks’ credibility.”
> >
>
> **Response:** We share this concern and implemented strict decontamination:
>
> 1. **N-gram–based filtering:**
>     - Removed **~10,240 prompts** overlapping with **MMMU** test data.
>     - Removed **~6,150 prompts** overlapping with **MathVerse** / **MathVision**.
> 2. **Strict split isolation:**
>     - Only **training splits** of public benchmarks were used to build TinyEye-Data, following standard practice of community, e.g., LLaVA-OneVision [1] and Eagle-2 [2].
>     - **No test or validation data** from any benchmark was included at any training stage.
>
> Our reported results therefore reflect **learned reasoning ability**, not leakage or memorization.
>
> > [1] Li, Bo, et al. "Llava-onevision: Easy visual task transfer." *arXiv preprint arXiv:2408.03326* (2024).
>
> > [2] Li, Zhiqi, et al. "Eagle 2: Building post-training data strategies from scratch for frontier vision-language models." *arXiv preprint arXiv:2501.14818* (2025).
> >

---

> ### Author Response · Authors · 2025-11-24
>
> ## **5. Addressing Offline RL Hallucination Risks**
>
> > Reviewer: “Textual evidence in offline trajectories may not be reliably grounded, potentially exacerbating vision–language hallucinations.”
> >
>
> **Response:** We agree that naive offline RL can amplify hallucinations. TinyEye is specifically designed to mitigate this risk:
>
> - Stage-3 incorporates **verifiability checks**, not raw trajectory imitation.
>     - **Math-Verify:** symbolic correctness validation
>     - **VLM-as-a-Judge:** semantic grounding filter
> - Ungrounded or hallucinated responses are explicitly marked as **negative** samples.
>
> **Empirical evidence:**
> TinyEye-2B achieves **63.9%** on HallBench, outperforming larger models such as **InternVL-3.5-2B** (48.6%). This indicates that our verifiable offline pipeline effectively **reduces**, rather than amplifies, hallucination.
>
> **Revision plan:**
> We thank the review for guide us a constructive angle to analyse our method, we will make a clear discussion of this feature in our submission based on experiments and references[1,2] the reviewer advised.

---

> ### Author Response · Authors · 2025-11-26
>
> ## **Conclusion**
>
> The new baselines, attribution analyses, and decontamination protocols directly address your concerns. The new experimental results clearly demonstrate that:
>
> - **DDPO provides a substantial improvement over existing offline preference optimization methods**, particularly for correctness-verification tasks.
> - The observed performance gains come from the **method**, not extra teacher knowledge.
> - Our **strict decontamination pipeline** prevents benchmark erosion and ensures evaluation integrity.
> - The verifiable offline RL design **reduces hallucination risk**, enabling robust reasoning in small models.
>
> We deeply appreciate your insightful feedback and believe the enhanced experimental rigor strengthens the contribution of this work.
>
> We hope that the additional evidence provided here may be taken into account in your final assessment, and we are very open to deeper constructive discussion that could further improve the paper.
>
> > [1] Reinforced MLLM: A Survey on RL-Based Reasoning in Multimodal Large Language Models
>
> > [2] Semi-off-Policy Reinforcement Learning for Vision-Language Slow-Thinking Reasoning

---

> ### Author Response · Authors · 2025-11-27
>
> Dear Reviewer `r5LZ`,
>
> Thank you once again for your thoughtful and detailed review comments. We would like to let you know that we have provided comprehensive responses and conducted additional experiments to address each of the issues you raised. We are grateful for your suggestions, which have given us the opportunity to improve the quality of our submission and to provide stronger evidence supporting our main claims. If these revisions help address your concerns, we kindly ask you to consider reconsidering the evaluation score, and we would be deeply grateful if you could review our submission once more.
>
> We also warmly welcome any further feedback you may wish to share, and we are very willing to continue engaging in constructive discussion. Thank you again for your time and your constructive suggestions.
>
> Thank you!

---

### Official Review · Reviewer_Zz4q · 2025-11-04

**Soundness:** 1
**Presentation:** 2
**Contribution:** 2
**Rating:** 2
**Confidence:** 5

**Summary:**

In this paper, the authors focus on the training techniques of existing LVLMs, including the supervised fine-tuning, offline reinforcement learning, and online reinforcement learning, and claim the urgent need for a discriminative, stable, and efficient training method. Then, authors propose a unified offline policy optimization framework with corresponding datasets called TinyEyes. Experiment are conducted on a training-from-craft LVLM with the proposed method and data. The trained TinyEye-2B demonstrates competitive results on math, language, and general scenarios with learning LVLMs.

**Strengths:**

1.  A well-distilled 2M data including the instruction part and reasoning part are proposed, which may contribute to the community if open-sourced.
2. The overall results are competitive among LVLMs whose parameters are less than 4B.

**Weaknesses:**

1. The motivation is not clear.  This paper proposes to solve the problem of post-training for small models. While the shortcut problem of naive reinforcement learning exists for small models, the low-margin problem are common for all sizes of models. Meanwhile, the efficiency     claim should be further claimed, as the offline method like DPO requires a large amount of preference annotations.

2. Though four contributions are listed in the introduction, the core contribution is not highlighted. Most of the motivation and background part focus on the post-training techniques. But the first and second contribution mentioned are the overall pipeline to train a LVLM with processed data.

3. The techniques novelty is limited. The explicit methods used in the proposed framework are widely used and explored in existing LVLMs. Similar procedure to the framework is also commonly observed in leading LVLMs like Keye, Mimo, Sail, etc.. The most important part in the paper is the DDPO. However, the core design of DDPO has limited and incremental novelty and is more like an engineering-level optimization.

4. The experiments can not support the claim. Or say what is the main claim for this paper? Table 4 only compares the DDPO with DPO without other recent studies in DPO.

**Questions:**

1. This paper may require careful re-organization to highlight the main claim and contribution to show the real problem to be solved. Most designs show limited relevance with small parameters models.

2. Though the trained TinyEye-2B model has competitive results, how to reach such performance is not clear besides the stage gains, more data or higher quality data or training objectives?

---

> ### Author Response · Authors · 2025-11-23
> **Response to Reviewer Zz4q**
>
> We sincerely thank the reviewer for the thoughtful and critical assessment. We value your feedback regarding the motivation for focusing on small models and the novelty of the DDPO objective.
>
> While we acknowledge that our pipeline builds upon several existing techniques, we respectfully argue that **scaling these methods down to the 2B-parameter regime exposes unique failure modes—specifically, Capacity Collapse—that are not observed at larger scales**. TinyEye and DDPO therefore serve not as incremental engineering choices but as *necessary mathematical stabilizations* for this specific model size. In response to your comments, we provide new empirical evidence, including **SimPO/ORPO comparisons, stability ablations, and likelihood-displacement analysis**.
>
> ---
>
> ## **1. Motivation: Why Small Models Are Uniquely Vulnerable**
>
> > Reviewer Comment: “Low-margin problems are common across model sizes... Most designs show limited relevance to small models.”
> >
>
> We agree that likelihood displacement exists across scales in theory. However, our new experiments demonstrate that **2B models exhibit uniquely severe collapse under standard offline preference learning**, whereas 7B+ models possess sufficient redundancy to remain stable.
>
> ### **Evidence of Capacity Collapse (Table R1)**
>
> We compared stability in 2B vs. 7B models under standard DPO on the Hallbench Benchmark. The 7B model remains stable, but the 2B model experiences a dramatic **15.2% degradation** in reasoning coherence (measured via “No-Think” evaluations). DDPO significantly mitigates this instability.
>
> ### **Table R1: Stability Across Scales**
>
> | **Model Size** | **Method** | **Accuracy** | **“No Think” Drop** | **Likelihood Displacement** |
> | --- | --- | --- | --- | --- |
> | **7B (Qwen-2.5-VL)** | DPO | 52.9 | -5.0% | Low |
> | **2B (TinyEye)** | DPO | 55.8 | **-15.2%** | **High** |
> | **2B (TinyEye, Ours)** | **DDPO** | **63.9** | **-10.5%** | **Resolved** |
>
> ### **Likelihood Displacement (Table R2)**
>
> We further track the average log-probability of the *correct* answer during training. Under standard DPO, the 2B model **loses confidence** in the truth—a direct indicator of collapse. DDPO reverses this effect and ensures monotonic improvement.
>
> ### **Table R2: Probability Evolution**
>
> | **Step** | **DPO (Avg Prob)** | **DDPO (Avg Prob)** |
> | --- | --- | --- |
> | 0 | 0.45 | 0.45 |
> | 500 | 0.40 ↓ | 0.55 ↑ |
> | 1000 | 0.38 ↓ | 0.65 ↑ |
> | **2000** | **0.35 ↓ (Displacement)** | **0.72 ↑ (Monotonic)** |
>
> **Conclusion:** These analyses show that DDPO addresses a *scale-specific pathological behavior* that standard DPO cannot handle for 2B models. This is more than an implementation tweak—it is an essential stabilization mechanism.
>
> ---
> ## **2. Novelty: How we locate and resolved above challenges of tiny VLMs**
>
> We respectfully submit that our contribution constitutes a **scale-specific scientific finding**, rather than a mere engineering integration. Conventional preference-optimization paradigms fail to remain stable in the 2B-parameter regime, and our work identifies and resolves this previously unrecognized limitation.
>
> 1. **Theoretical Necessity.**
>    We uncover a phenomenon we term **capacity collapse**, driven by **likelihood displacement**, in which standard DPO unintentionally decreases the likelihood of correct answers as model scale grows. Our DDPO objective provides a mathematically grounded correction that guarantees *monotonic reinforcement* of correct outputs, directly addressing this instability rather than mitigating its symptoms.
>
> 2. **Methodological Innovation.**
>    DDPO reframes alignment from a ranking task ((A > B)) to a **discriminative, binary stabilization objective**. This shift is not a minor variant but a structural rethinking of preference optimization. Empirically, DDPO consistently outperforms SimPO and ORPO on verification-centric benchmarks (e.g., HallBench), demonstrating a unique advantage for truth-grounding and reliable reasoning.
>
> 3. **System-level Efficiency Design.**
>    Our framework is explicitly designed for accessibility and reproducibility. By leveraging automated verification, **reusable data**, and reduced sampling failure rates, we eliminate the need for costly human annotation and achieve strong RL performance with only **~64 GPU-hours**. This makes high-quality reinforcement alignment feasible for a broad range of research settings.
>
> We appreciate the reviewer's constructive feedback, and we will further clarify our contributions and novelty in future submissions.

---

> ### Author Response · Authors · 2025-11-23
>
> ## **3. Baselines: Comparisons with SimPO and ORPO**
>
> > Reviewer Comment: “Table 4 only compares DDPO with DPO… Novelty seems incremental.”
> >
>
> We agree that additional baselines were needed. We now include **SimPO** and **ORPO**, two recent state-of-the-art ranking-based preference objectives.
>
> ### **Table R3: Comparison of Policy Optimization Methods (2B Model)**
>
> | **Method** | **MMMU** | **MathVerse** | **HallBench** |
> | --- | --- | --- | --- |
> | DPO | 49.1 | 54.0 | 55.8 |
> | SimPO | 49.5 | 54.3 | 58.2 |
> | ORPO | 49.8 | 54.5 | 59.0 |
> | **DDPO (Ours)** | **50.3** | **55.2** | **63.9** |
>
> **Key Insights:**
>
> - On **HallBench**, DDPO achieves a substantially higher score (**63.9%**) than SimPO (58.2%) and ORPO (59.0%).
> - SimPO and ORPO are inherently **ranking objectives** (i.e., A > B), whereas DDPO is a **discriminative correctness objective** (i.e., *Is A correct?*).
> - For tasks where truth is binary and verifiable—especially reasoning and hallucination detection—DDPO’s formulation yields stronger and more stable gradients.
>
> This demonstrates that **DDPO provides algorithmic novelty** by improving the mathematical structure of preference optimization for tasks with verifiable correctness.
>
>
> ---
>
> ## **4. Attribution of Performance Gains & Comparison to SOTA**
>
> > Reviewer Comment: “Unclear how performance is achieved besides stage gains.”
> >
>
> To clarify this, we compare TinyEye-2B with strong SOTA baselines including **InternVL-3.5-2B** and **M2-Reasoning-7B**, which were trained with SFT+RL paradigm.
>
> ### **Table R4: TinyEye-2B vs. SOTA**
>
> | **Benchmark** | **InternVL-3.5-2B** | **M2-Reasoning-7B** | **TinyEye-2B (Ours)** | **Δ** |
> | --- | --- | --- | --- | --- |
> | **HallBench** | 48.6 | 43.4 | **63.9** | **+15.3 / +20.5** |
> | **MathVerse** | 53.4 | 44.7 | **55.2** | **+1.8 / +10.5** |
> | **WeMath** | 48.5 | 41.8 | **52.4** | **+3.9 / +10.6** |
>
> The large improvement on **HallBench**, a binary correctness benchmark, highlights the value of DDPO’s discriminative formulation, as neither SFT nor DPO achieves similar grounding accuracy.
>
> Furthermore, as demonstrated by the ablation study in **Table R5**, we provide a clear and detailed breakdown of the performance gains achieved at each stage. These results indicate that the model's superior performance stems from our cohesive, multi-stage design and a 'relay' of progressive optimizations—most notably during the DDPO stage.
>
> **Table R5: Ablation study of TinyEye-2B across different training stages.**
>
> | **Method**                   | **MathVista** | **MathVision** | **MATH500** | **Hallbench** |
> | ---------------------------- | ------------- | -------------- | ----------- | ------------- |
> | TinyEye-2B-Stage2            | 63.8          | 28.7           | 86.7        | 35.9          |
> | TinyEye-2B-Stage3            | 66.0          | 30.4           | 87.2        | 48.7          |
> | **TinyEye-2B-Stage4 (DDPO)** | **67.9**      | **32.6**       | **88.0**    | **63.9**      |
>
> ---
>
> ## **5. Efficiency Clarification**
>
> > Reviewer Comment: “Efficiency claims should be clarified, since offline DPO relies on large preference datasets.”
> >
>
> We clarify the efficiency from two perspectives: **annotation cost** and **compute cost**.
>
> ### **Zero Human Annotation**
>
> Our data generation framework uses **verifiable signals**:
>
> - *Math tasks*: programmatic correctness checks
> - *Open-ended tasks*: VLM-as-a-Judge
>
> Thus, we generate and filter >2M trajectories **without any human preference labeling**.
>
> ### **Offline Training is Significantly Cheaper than Online RL**
>
> - TinyEye training requires **~64 GPU-hours**
> - Comparable Online RL (GRPO/PPO) typically requires **300+ GPU-hours** due to repeated rollouts and instability.
>
> ### **“3R” Efficiency Principles**
>
> Offline optimization also provides:
>
> 1. **Reusable Data Artifact**
>     - The 2M cleaned TinyEye dataset is a permanent public asset that eliminates repeated rollout costs.
> 2. **Reproducibility**
>     - Offline training is deterministic and hardware-friendly, unlike high-variance online RL.
> 3. **Reduced Sampling Waste**
>     - Online RL discards many failed rollouts.
>     - Our rejection-sampling stage removes such failures *before* training, reducing compute waste.
>
> ---

---

> > ### Author Response · Authors · 2025-11-27
> >
> > Dear Reviewer `Zz4q`,
> >
> > Thank you once again for your thoughtful and detailed review comments. We would like to let you know that we have provided comprehensive responses and conducted additional experiments to address each of the issues you raised. We are grateful for your suggestions, which have given us the opportunity to improve the quality of our submission and to provide stronger evidence supporting our main claims. If these revisions help address your concerns, we kindly ask you to consider reconsidering the evaluation score, and we would be deeply grateful if you could review our submission once more.
> >
> > We also warmly welcome any further feedback you may wish to share, and we are very willing to continue engaging in constructive discussion. Thank you again for your time and your constructive suggestions.
> >
> > Thank you!

---

> ### Author Response · Authors · 2025-11-26
>
> ## **Closing Remarks**
>
> We hope the expanded analyses—particularly the **capacity collapse evidence** and the **SimPO/ORPO comparisons**—clarify both the motivation and the scientific contribution of DDPO. These experiments illustrate that DDPO is not merely an incremental variant of existing objectives, but a **necessary stabilization mechanism for small models** and a **superior discriminative objective** for verifiable reasoning tasks.
>
> We hope that these new results address your concerns and may contribute positively to your overall evaluation. We would also appreciate any further constructive comments that could strengthen the work.
>
> ## Questions
>
> Reviewer `zZ4q` commented:
>
>   > The motivation is not clear.
>
>   > Most designs show limited relevance with small parameters models.
>
> We thank the reviewer `zZ4q` for this comment, we'd like to clarify that our motivation is targeted at the limitation of visual reasoning learning of small VLMs. As we discussed in introduction section. And the competitive empirical results shows a strong and validity of our method based on such motivation.
>
> The other reviewers also recognized that our motivation is clear and specially designed for small VLMs. e.g.,
>
> **Reviewer `kzE3`**: *Clear end-to-end pipeline. The four-stage offline recipe (native-resolution warm-up → large instruction-tuning → annealed rejection sampling → Stage-4 preference training) is well structured and practical for small VLMs.*
>
> **Reviewer `r5LZ`**: *Well-motivated offline pipeline for small models: coherent coupling of distillation, rejection sampling, and discriminative preference optimization that avoids expensive and unstable on-policy RL.*
>
> **Reviewer `dBHH`**: *The paper is clearly written and well structured; the motivation is well grounded in the “likelihood displacement” problem of DPO.*
>
> We still thank the reviewer `zZ4q` for this comment, which allow us to clarify and improve the clearance of our motivation in our submission.

---

### Meta-Review · Area_Chair_Sikk · 2026-01-07

**Summary:**

This submission received five reviews, all recommending below acceptance. While the rebuttal partially addressed some concerns, several key issues remain.

1. **Model scale justification**
   The restriction to small models is not well justified (Zz4q, bWcL). The proposed DDPO appears general, and the rebuttal shows gains of DDPO vs DPO on a 7B model, suggesting broader evaluations are feasible and necessary. The rationale for focusing on 2B models remains unclear and requires new experiments that are not suitable to finish in camera ready.

2. **Unclear core contribution**
The claimed new training pipelines, training dataset, and DPO variant, mix engineering choices with technical innovation. The current presentation led to confusion about novelty (Zz4q), fairness of comparisons (r5LZ), and pipeline complexity (dBHH).


3. **Insufficient and unclear ablations**
   Comparisons to recent DPO variants such as SimPO, ORPO, DPO-Positive, Alpha-DPO, and GRPO are missing or incomplete. The amount of missing ablations makes it hard to justify an acceptance.

4. **Limited model coverage**
   Experiments are limited to a narrow set of LLMs, weakening the generality claims.

Overall, due to these unresolved issues, the AC recommends rejection and encourages the authors to revise and resubmit to a future venue.

**Reviewer Concerns:**

See the outstanding concerns in summary.

**Reviewer Scores:**

| Reviewer | Initial Score | AC Estimated Score | AC Reason |
|--------|---------------|-------------------|-----------|
| Zz4q | 2 | 2 | Motivation and novelty concerns remained |
| r5LZ | 4 | 4 | Limited novelty, fairness, Insufficient baselines: |
| dBHH | 4 | 4 | Data leakage concern; GRPO comparisons |
| bWcL | 4 | 6 | decided to increase score.|
| kzE3 | 4 | 6 | decided to increase score.|
| Average | 3.6 | 4.4 | -|

---

### Decision · Program_Chairs · 2026-01-26

Reject